# The Centriolar Adjunct–Appearance and Disassembly in Spermiogenesis and the Potential Impact on Fertility

**DOI:** 10.3390/cells8020180

**Published:** 2019-02-19

**Authors:** Anastasiia S. Garanina, Irina B. Alieva, Elizaveta E. Bragina, Emmanuelle Blanchard, Brigitte Arbeille, Fabrice Guerif, Svetlana Uzbekova, Rustem E. Uzbekov

**Affiliations:** 1Faculté de Médecine, Université de Tours, 10, Boulevard Tonnellé, 37032 Tours, France; anastasiacit@gmail.com (A.S.G.); emmanuelle.blanchard@univ-tours.fr (E.B.); brassart@univ-tours.fr (B.A.); fabrice.guerif@univ-tours.fr (F.G.); 2National University of Science and Technology «MISiS», 119049 Moscow, Russia; 3Belozersky Institute of Physico-Chemical Biology, Moscow State University, Leninskye gory 1-40, 119992 Moscow, Russia; irina_alieva@mail.ru (I.B.A.); bragor@mail.ru (E.E.B.); 4Research Centre for Medical Genetics, Moskvorechye 1, 115478 Moscow, Russia; 5CHRU, UF Biologie Cellulaire & Microscopie Electronique, 2, Boulevard Tonnellé, 37032 Tours, France; 6INSERM U1259, 10, Boulevard Tonnellé, 37032 Tours, France; 7UFR de Sciences et Techniques, Physiologie de la Reproduction et des Comportements (PRC), INRA, Centre de Tours, 37380 Nouzilly, France; svetlana.uzbekova@tours.inra.fr; 8Service de Médecine et Biologie de la Reproduction, CHRU Bretonneau, 2, boulevard Tonnellé, 37044 Tours, France; 9Faculty of Bioengineering and Bioinformatics, Moscow State University, Leninskye gory 73, 119992 Moscow, Russia

**Keywords:** centriolar adjunct, centriole, centrosome, idiopathic male sterility (IMS), human spermatozoa ultrastructure, in vitro fertilization (IVF)

## Abstract

During spermiogenesis, the proximal centriole forms a special microtubular structure: the centriolar adjunct. This structure appears at the spermatid stage, which is characterized by a condensed chromatin nucleus. We showed that the centriolar adjunct disappears completely in mature porcine spermatozoa. In humans, the centriolar adjunct remnants are present in a fraction of mature spermatids. For the first time, the structure of the centriolar adjunct in the cell, and its consequent impact on fertility, were examined. Ultrastructural analysis using transmission electron microscopy was performed on near 2000 spermatozoa per person, in two patients with idiopathic male sterility (IMS) and five healthy fertile donors. We measured the average length of the “proximal centriole + centriolar adjunct” complex in sections, where it had parallel orientation in the section plane, and found that it was significantly longer in the spermatozoa of IMS patients than in the spermatozoa of healthy donors. This difference was independent of chromatin condensation deficiency, which was also observed in the spermatozoa of IMS patients. We suggest that zygote arrest may be related to an incompletely disassembled centriolar adjunct in a mature spermatozoon. Therefore, centriolar adjunct length can be potentially used as a complementary criterion for the immaturity of spermatozoa in the diagnostics of IMS patients.

## 1. Introduction

Several important morphological changes occur within the spermatid during the last part of spermiogenesis, more precisely during their transformation into mature sperm cells.

One of these is chromatin compactization. The chromatin of morphologically mature spermatozoa looks like a homogeneous electron-dense mass. By contrast, the chromatin from abnormal spermatozoa presents with granular material that is typical for spermatids at the stage of nuclei elongation [1,2,3]. The term “immature chromatin” describes this kind of spermiogenesis anomaly. This chromatin morphology correlates with a high content of testis-specific histones, which are not replaced by protamines [1,4], and with chromosome aberrations [5]. In the spermatozoa of patients with idiopathic infertility, electron microscopic investigation reveals a significant increase in the number of spermatozoa with immature chromatin [6]. The latter was shown to significantly decrease fertilization ability [7].

The effects of various ultrastructural centriole abnormalities in mature spermatozoa upon male fertility have not been thoroughly studied. Little is known about the relationship between various ultrastructural centriole abnormalities in mature spermatozoa and fertilization ability. The proximal centriole of spermatozoa is duplicated after fertilization prior to zygote cleavage, and participates in the formation of one of the poles in the first mitotic spindle [8,9]. The second pole of the zygote spindle can be organized by the activity of the atypical centriole, which originates from the morphologically degenerated during spermiogenesis distal centriole of spermatid [10,11].

During spermiogenesis, the proximal centriole forms a special microtubular structure: the centriolar adjunct (centriole microtubular extention in old terminology) [12]. This structure appears at the spermatid stage, which is characterized by a condensed chromatin nucleus, and may attain a length two or three times that of the proximal centriole. It has been identified in all mammalian species examined thus far [13]. Surprisingly, the exact function of this organelle is still unknown, and there has been no recent research on it; the last publications are more than 15 years old [14,15,16]. The centriolar adjunct demonstrates dynamic changes in size, decreasing during spermatozoa maturation, and disappearing partially or completely in the mature spermatozoa of different organisms [12,13,17,18,19]. In particular, the complete disassembly of the centriolar adjunct has been shown in rabbit spermatozoa, and almost complete disassembly of the centriolar adjunct has been found for *Macaca nemestrina* mature sperm cells [18]. 

On the other hand, the preservation of relatively short and partially disorganized microtubule triplets, doublets, and singlets of the centriolar adjuncts has been shown in human spermatozoa [3,17,18]. It has been suggested that the presence of a centriolar adjunct in human sperm cells can be considered a manifestation of incomplete maturation [18]. 

Thus, disturbances in proximal centriole morphogenesis, in particular incomplete disassembly of the centriolar adjunct, can be one of the reasons for embryonic development arrest at the zygote stage. 

In this study, using transmission electron microscopy (TEM), we compared the ultrastructure of chromatin and the centriolar complex of spermatozoa from infertile patients (repeated zygote arrest after in vitro fertilization, or IVF) and fertile donors.

Samples of porcine testicular and sperm cells were used to illustrate the disassembly of the centriolar adjunct during last part of spermiogenesis: the transition from spermatid to mature spermatozoon.

The aim of this study was to investigate a cause for the infertility of morphologically normal spermatozoa, leading to the development of zygote arrest at the first division stage after IVF, using detailed ultrastructural analysis of the sperm from idiopathic male sterility (IMS) patients. This study suggests, for the first time, that disassembly of the centriolar adjunct is required for the functional maturity of human spermatozoa. However, it must be noted that other mechanisms are possible, and the centriolar adjunct may be a pleiotropic phenotype. 

## 2. Materials and Methods

### 2.1. Human Sperm Analysis

Biological material was obtained from the two unfertile patients with idiopathic male sterility (IMS) and five fertile sperm donors. All patients and donors gave their written consent to use the samples of their biological material for research purposes.

Patient #1 (P1), 41 years old, was from a couple that performed three unsuccessful IVFs after seven failed sperm inseminations. After the third IVF failure, P1’s sperm was collected for sperm ultrastructure analysis with transmission electron microscopy (TEM). The main characteristics of the sperm were normal. P1’s wife had no phenotypes that could contribute to infertility. Both male and female karyotypes were normal.

Patient #2 (P2), 34 years old, was from a couple that had no pregnancies after two years of unprotected intercourse, and then two unsuccessful IVFs. After the second IVF failure, P2’s sperm was collected for ultrastructure analysis by TEM. The main characteristics of the sperm were normal. Both male and female karyotypes were also both normal. Cryopreserved fertile donor semen was used later for insemination. A healthy baby girl was delivered at 40 weeks of gestation. 

For a comparative analysis of centriolar complex structures, normal mature spermatozoa were obtained from five healthy fertile donors (D1–D5) and used as a control. 

### 2.2. Porcine Testicles and Sperm Analysis

Sperm collection procedure on animals (*Susscrofa domesticus*) was conducted in accordance with the guidelines for the care and use of laboratory animals issued by the French Ministry of Agriculture. The study was performed in our experimental pigsty (Unité Expérimentale de Physiologie Animale de l’Orfrasière, UEPAO, Nouzilly, France) with the agreement of Indre-et-Loire Service of Animal Health and Protection (Direction Départementale de la Chohésion Sociale et de la Protection des Populations, DDCSPP) under the number SA 1500878. Porcine testicle samples were obtained from adult boars at a local commercial slaughterhouse; no experiments were performed on live animals.

### 2.3. Transmission Electron Microscopy (TEM)

Human spermatozoa, porcine spermatozoa, and small pieces (10 × 1 × 2 mm) of porcine testicle were fixed in 2% glutaraldehyde (Agar Scientific, France) solution in 0.1 M of cacodylate (Merck, Hohenbrunn, Germany) buffer (pH 7.4) for one hour, washed three × 10 min in 0.1 M of cacodylate buffer, and post-fixed for one hour with 2% osmium tetroxide (Electron Microscopy Science, Hatfield, PA, USA) in 0.1 M of cacodylate buffer. After washing in 0.1 M of cacodylate buffer for 10 min and two × 10 min in distillated H_2_O, samples were dehydrated in a graded series of ethanol solutions (50% ethanol two × 10 min; 70% ethanol three × 15 min; 90% ethanol three × 20 min; and 100% ethanol three × 20 min). Final dehydration was performed by 100% propylene oxide (PrOx, TermoFisher GmbH, Kandel, Germany) three × 20 min. Then, samples were incubated in PrOx/EPON epoxy resin (Sigma-Aldrich, St. Louis, MO, USA) mixture in a 3:1 ratio for two hours with closed caps, 16 h with open caps, and in 100% EPON for 24 h at room temperature. Samples were replaced in new 100% EPON and incubated at 37 °C for 48 h and at 60 °C for 48 h for polymerization. 

Ultra-thin sections (thickness 70 nm) were cut with a “Leica Ultracut UCT” ultramicrotome (Leica Microsysteme GmbH, Wien, Austria), stained with 5% uranyl acetate (Electron Microscopy Science, Hatfield, PA, USA), and placed on TEM one-slot grids coated with Formvar film. The sections were then observed at 100 kV with a Jeol 1011 TEM (Tokyo, Japan) connected to a Gatan digital camera driven by Digital Micrograph software (GMS 3, Gatan, Pleasanton, CA, USA).

### 2.4. Measurements and Statistical Analysis

Longitudinal sections of the “proximal centriole + centriolar adjunct” complexes were used to measure their length and diameter by TEM. The diameter was also measured on the cross-sections.

Photos were processed in Adobe Photoshop CS3 software, and statistical analysis was performed in Excel 2007 software. For statistical comparisons, analyzed t-tests were performed; *p* values of < 0.01 were considered statistically significant. 

### 2.5. Three-Dimensional Reconstruction 

We recently described the use of serial TEM sections for three-dimensional (3D) reconstruction [20,21]. We used a similar approach to generate a 3D reconstruction of the centriole–centriolar adjunct complex in spermatozoa. Photoshop CS3 software was used to align images from consecutive serial ultra-thin section (70-nm thick) stacks. Contours were drawn with IMOD (Image MODel) software (version IMOD 4.1) and then arranged into objects. The contours of each object were then joined, using the IMODmesh feature, to form a 3D model. 

## 3. Results and Discussion

The centriolar adjunct is a unique microtubules-based organelle of mammalian cells, which is connected with the proximal centriole of spermatids [12]. It is not detected in spermatids until such time as the flagellum starts to grow from the distal centriole. Here, we used porcine spermatids (*Susscrofa domesticus*) to analyze the structure of the centriolar adjunct. 

### 3.1. Centriolar Adjunct Structure in Pig Spermatids

The centriolar adjunct appears in spermatids with condensed chromatin nucleus in parallel with distal centriole disappearance (Figure 1). The centriole contains nine triplets of MTs (microtubules) (Figure 1e), and the flagellum contains nine doublets of MTs with two central MTs (Figure 1g). Cross-sections of the centriolar adjunct show that the morphology of this structure significantly differs from the centriole and flagellum (Figure 1f). The wall of the centriolar adjunct consists of electron-dense material with nine complete “A” MTs and 18 incomplete “B” and “C” MTs immersed in it. This structure of the centriolar adjunct cross-sections coincides well with that described earlier for centriolar adjuncts in chinchilla [13] and man [18]. Only the degree of openness for MT “B” and “C” can differ somewhat, but this may be due to a discrepancy in the level at which the cross-section of the centriolar adjunct was made. The internal volume of the centriolar adjunct has a periodic, rather than a strictly regular, system of dense disks along the whole structure (Figure 1a). Similar periodic electron-dense constrictions of the inner lumen of the centriolar adjunct in spermatids of human [18] and goat [19] were shown previously. Thus, the internal structure of the centriolar adjunct described by us in porcine spermatids was almost identical to the previously described structure of the centriolar adjunct in human spermatids [18]. 

The function of the centriolar adjunct is still unknown. Hypothetically, in our view, it may be important for sensitivity functions, or may protect the proximal centriole from disassembly. It was suggested that a centriolar adjunct can play a role in determining the plane of the nucleus flattening (the mechanism of this “regulation” is unclear) [19]. Similar to the spermatids nucleus of a goat, porcine spermatids had flattened ovoid nuclei. We can confirm the data obtained in work [19] that the position of the centriolar adjunct was not accidental; together with the flagellum, it formed a plane that was parallel to the plane of the spermatid nucleus flattening (N = 10). In mature porcine spermatozoa, the centriolar adjunct disappears (Figure 1h–j), as has been shown before for many other animals [13,17,18,19]. So, it seems that this spermatozoa organelle is not necessary for early centriole development in zygotes and during first embryo divisions. On the contrary, we hypothesize that incomplete centriole adjunct disassembly could be a reason for some development dysfunctions. 

### 3.2. Centriolar Adjunct Structure in Human Spermatozoa

Exploring the spermatozoa of healthy donors (as controls) and IMS patients, we found that in some cells, the proximal centrioles were associated with structures similar to centriolar adjuncts, which we describe in porcine spermatids (Figure 1 and Figure 2). Unlike the latter, there was no electron-dense material in the walls and no discs in the internal volume of human spermatozoa centriolar adjuncts. Moreover, the latter had an irregular structure with doublets and singlets of MTs as well as incomplete MTs. At the same time, detailed ultrastructural analysis revealed that the proximal centriole had a normal structure in both the longitudinal (Figure 2a) and transverse (Figure 2d) sections.

The border between the centriolar adjunct (CA) and the proximal centriole (PC) was not obvious, so we measured the average length of “the proximal centriole + the centriolar adjunct” (PC+CA) complex. 

The length of this complex in the spermatozoa of donors was shorter than in the spermatozoa of patients, because their CA was partially or completely disassembled (Figure 2 and Appendix A). The average length of the PC+CA complex in the spermatozoa of the fertile donors was 558 ± 137 nm (n = 150, n = 30 for each of the five donors). The lengths were consistent between donors, with a range from 543 ± 111 nm to 569 ± 162 nm; the *t*-test-derived *p*-value was between 0.45–0.9. This data show that there are no significant differences between all the donors in this parameter. 

At the same time, it should be noted that in contrast to porcine spermatozoa, where the PC were of similar length (Figure 1d), the variation of PC+CA complex length in the spermatozoa of fertile donors was very high (Figure 2e). Therefore, we performed a comparative analysis of the PC+CA lengths for each class: class I—PC without CA, between 300–499 nm; class II—PC with short CA, between 500–699 nm; class III—PC with medium-length CA, between 700–899 nm; class IV—PC with long CA, between 900–1099 nm; and class V—PC with very long CA, between 1100–1299 nm (Figure 3, Appendix A). In total, near 2000 spermatozoids per person were analyzed by TEM. The percentage of centrioles with a very visible centriolar adjunct in the spermatozoa of both patients was more than 90%; in the spermatozoa of all five donors, it was about 50%. Only the spermatozoids in which proximal centrioles had parallel orientation to the section plane were used for quantitative analysis of PA+PC length (n = 30 per each patient and for each donor). 

The mean length of PC+CA complexes was 1059 ± 100 nm in P1 spermatozoa (n = 30) and 769 ± 160 nm in P2 spermatozoa (n = 30), which represented a significant difference between the two patients (*p* = 3.75 × 10^−8^). The PC+CA lengths of both IMS patients were significantly longer than those of the donors (*p* values: 4.99 × 10^−20^–2.52 × 10^−6^; Figure 2e).

The majority of spermatozoa from fertile donors had PC+CA lengths within classes I and II. (Figure 2f, Figure 3 and Appendix A), and their CAs were partially or completely disassembled (Figure 2a, Figure 3b, Appendix A, and Figure 3a, Appendix A, respectively). In previous studies, the lengths of PC and CA were not measured, and only short CAs were reported [15,16], similar to Figure 2a.

In contrast to fertile donors, for IMS patients, only a small part of the spermatozoa had a length of PC+CA less than 600 nm (classes I and II; Figure 2f). The majority of PC+CA complexes from P1’s spermatozoa were in the range found in classes IV and V (Figure 2f and Figure 3d,e; Appendix A), and those from P2 spermatozoa were in the ranges found in classes III and IV (Figure 2f,g; Figure 3c; Appendix A). 

All of the spermatozoids of fertile boar had proximal centrioles with a completely disassembled centriolar adjunct (Figure 1d,h–j). We can propose that the fertilization capacity is proportional to the ratio of spermatozoa with a fully or almost completely disassembled centriolar adjunct (PC+CA complexes length class I). For fertile donors 33.3–43.3% of spermatozoa were within this class, with an average for D1–D5 of 40% ± 4.7%. By contrast, patients with IMS had only 3.3% (P1) and 6.7% (P2) of spermatozoa within class I of PC+CA lengths. 

CA shortening and its role in maintaining fertility had not been investigated before this work. Based on the premises that nature does not allow anything superfluous and continuous natural selection would not retain such a complex organized structure if it did not have an important function for life, we decided to undertake our current study. Based on the structure of the CA, we propose that this structure might be partially analogous to another centriole derivate, the primary cilium, and in particular, the primary cytosolic cilia, which is not yet surrounded by a membrane at this early stage of formation [22]. 

From one side, similar to the primary cilium, the CA does not have central MTs, which suggests the stillness of the CA. From the other side, as opposed to the motile cilia, both primary cilia and CA are growing from the mother centriole [23]. It thought sometimes that the distal centriole, which forms the flagellum, is the mother centriole (similar to the mother centriole that forms the primary cilium in somatic cells); however, our data obtained on early porcine spermatids [23] showed that the daughter centriole forms a flagellum in sperm cells.

In recent years, the relationship between the primary cilium and the regulation of the cell cycle has been established [24,25]. The primary cilium appears in cells after the end of mitosis and disappears before the onset of a subsequent mitosis. Regulation of the length of the cilium is accomplished by at least three kinase-related mechanisms [24]. Moreover, it has been shown that disruption of the primary cilium disassembly leads to a block of mitosis. These data allowed us to hypothesize that the CA can also be a negative regulator of first mitotic divisions, zygote cleavages, or possibly the first few divisions of blastomeres. One can hypothesize that the malfunction of CA disassembly during spermiogenesis could be caused by the anomalies in regulatory kinase systems, and the CA might serve as a kind of” brake” on a faulty embryonic process. 

A detailed study of primary cilium disassembly dynamics during the cell cycle revealed that this process is regulated by various kinases, at different stages. In the first stage, disassembly of the primary cilium is regulated by Aurora-A kinase, while further disassembly depends on the Plk1 (polo-like kinase 1) and Nek2 (NIMA (Never In Mitosis)-Related Kinase 2) kinases [24]. Extrapolating these data to the shortening of CA, we hypothesized that several different mechanisms might participate in CA disassembly, and only the normal functioning of each one would allow the spermatozoid to act as the progenitor for a new organism. Otherwise, the development of the embryo would stop at an early stage, as observed in IMS patients. 

### 3.3. Chromatin State in Nuclei of Human IMS Patients and Donors’ Spermatozoa

Along with the presence of CA, another sign of the abnormal development of a spermatozoon is its incomplete nuclear chromatin condensation. These two morphological anomalies in spermatozoa are inherent in spermatids [1,2], and can indicate spermatozoa immaturity. The next question was whether these two types of deviations from the norm are related to each other. 

If they are connected, their individual analysis is not required for medical diagnosis. It will be enough to study the more easily detectible anomaly: the incomplete condensation of chromatin.

The data presented in Table 1 showed that the length of the PC+CA complexes did not correlate with the percentage of cells with incomplete chromatin condensation. The percentage of P1 spermatozoa with incomplete chromatin condensation was higher than in donor cells, while that of P2 was at the lower end of the range for D1–D5 (Table 1).

Additionally, all of the patients’ spermatozoa with detected PC+CA complexes were investigated for their nuclei chromatin state. We revealed all four possible variants of combinations: spermatozoa with condensed chromatins and long adjuncts, spermatozoa with condensed chromatins and disassembled (short) adjuncts, spermatozoa with incomplete condensed chromatins and long adjuncts, and finally, spermatozoa with incomplete condensed chromatins and disassembled adjuncts (Figure 4). 

The average length of the PC+CA complex in the P1 spermatozoa with condensed chromatin (1044.4 ± 127.8 nm) did not differ significantly (*t*-test; *p* = 0.60) from this parameter for spermatozoa with incomplete condensed chromatin (1013.4 ± 181.2 nm). Therefore, our results clearly show that defects in sperm maturation, which are expressed as incomplete chromatin condensation, and a lower disassembly of the centriolar adjunct, are independent anomalies of spermatozoa development. 

Thus, we can conclude that the disassembly of the centriolar adjunct is required for the functional maturity of human spermatozoa, and it does not depend on nucleus chromatin condensation. Both these parameters can be estimated only by TEM. Since these morphological abnormalities are independent, a decrease in fertility (the proportion of spermatozoa that are capable of fertilization and the formation of a normally developing embryo) for each of the parameters leads to a proportional decrease in total fertility. That is, if only 6% of P2 spermatozoa have a completely disassembled adjunct and 88.8% have a normal condensed chromatin, then the proportion of “normal spermatozoa” (only in these two parameters) will be 0.06 × 0.888 = 0.053. For comparison, the proportion of normal spermatozoa for P1 was only 0.033 × 0.399 = 0.013 (Table 1).

## 4. Conclusions

The deterioration of human reproductive health in developed countries leads to an increase in the number of couples who are incapable of natural conception [26]. One of the ways to overcome this problem is assisted reproductive technology (ART; e.g., artificial insemination or in vitro fertilization [IVF]), which involves the manipulation of gametes isolated from the body. However, these technologies are effective only in cases when the gametes that are used for fertilization are functionally normal. 

Despite the high efficiency of intracytoplasmic sperm injection (ICSI), embryologists often face zygote arrest on day two, despite apparently “normal” fertilization on day one. The tests that are routinely used to evaluate sperm characteristics include numeration, motility, vitality, and morphology; however, they remain unable to help explain potential sperm defects such as early developmental arrest. 

However, some defects in the development of sperm can be detected only in the course of electron microscopic examination. Consequently, in some cases, standard diagnostic methods for IMS are insufficient, and an ultrastructural analysis is required, which will make it possible to exclude useless in vitro fertilization attempts in the case of idiopathic male sterility caused by a violation in the ultrastructure of the centriolar complex.

This paper gives an explanation of one of the reasons for blocking the development of a zygote after in vitro fertilization, and transfers this case from the category of «idiopathic» to the category of diagnosable disorders of sperm development. 

## Figures and Tables

**Figure 1 cells-08-00180-f001:**
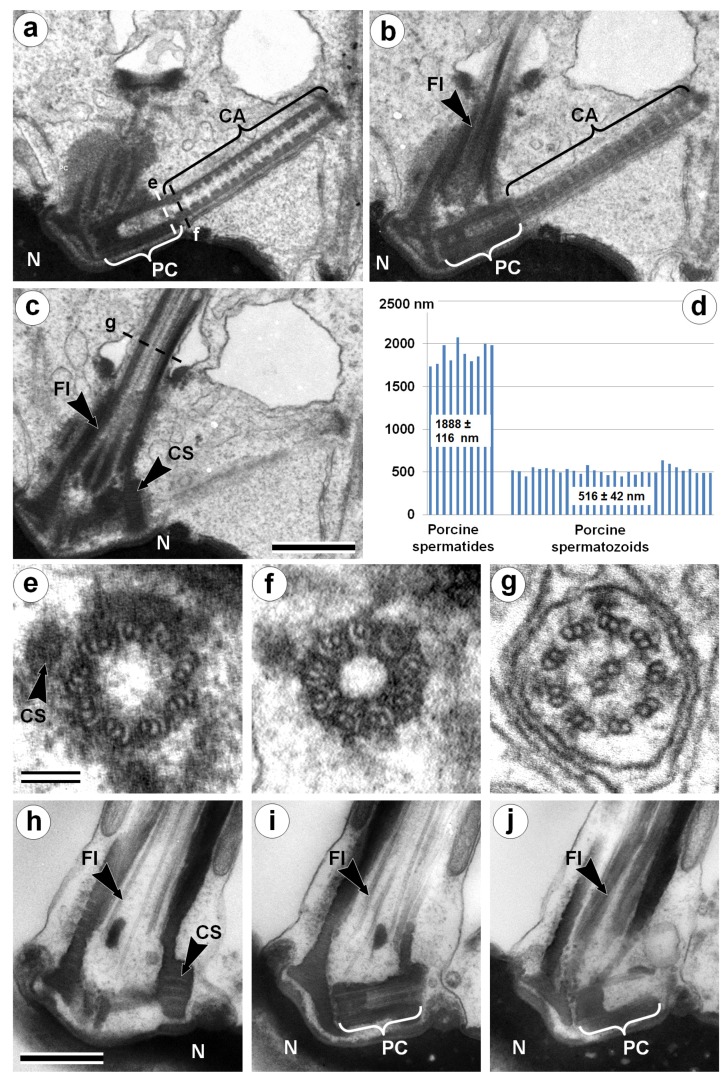
TEM analysis of the neck region of a porcine spermatid with condensed chromatin nucleus (**a**–**c**,**e**–**g**) and a mature spermatozoon (**h**–**j**). Three consecutive longitudinal ultra-thin sections of the centriolar adjunct and flagellum in the spermatid with a condensed chromatin nucleus (**a**–**c**); three cross-sections show the ultrastructure of the proximal centriole (**e**), centriolar adjunct (**f**), and flagellum (**g**). (**d**) Statistical analysis of the length of the proximal centriole (PC) + centriolar adjunct (CA) complex in porcine spermatids and mature spermatozoa. (**h**–**j**) Three consecutive longitudinal ultra-thin sections of the PC and flagellum in mature spermatozoon. CA: centriolar adjunct; CS: collar-like structure near distal part of proximal centriole; Fl: flagellum; N: nucleus; PC: proximal centriole. Scale bar in (**a**–**c**) and in (**h**–**j**) 500 nm; in (**e**–**g**) 100 nm.

**Figure 2 cells-08-00180-f002:**
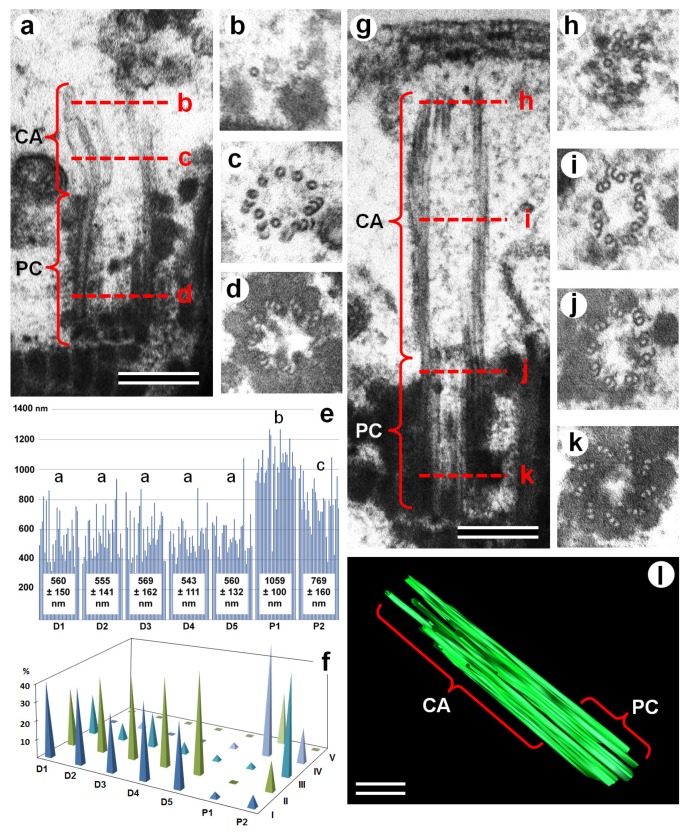
TEM analysis of the structure of the proximal centriole and centriolar adjunct in spermatozoa from healthy fertile donors (**a**–**d**) and IMS patients (**g**–**k**). (**a**,**g**) Longitudinal ultra-thin sections of PC+CA complexes; (**b**–**d**,**h**–**k**) cross-sections of the PC+CA complexes at different levels, as indicated by red dashed lines on images (**a**) and (**g**). (**e**,**f**) Histogram and length classes (I: 300–499 nm; II: 500–699 nm; III: 700–899 nm; IV: 900–1099 nm; and V: 1100–1299 nm) distribution of PC+CA complex lengths (in nm) in healthy fertile donors (D1–D5) and IMS patients (P1, P2); different letters mean significant differences between the data groups. (**l**) Three-dimensional (3D) reconstruction of PC+CA complex in an idiopathic male sterility (IMS) patient, showing the continuity of the microtubules (MTs) of the proximal centriole and centriolar adjunct. CA: centriolar adjunct; IMS: idiopathic male sterility; MTs: microtubules, PC: proximal centriole. Scale bar: 200 nm.

**Figure 3 cells-08-00180-f003:**
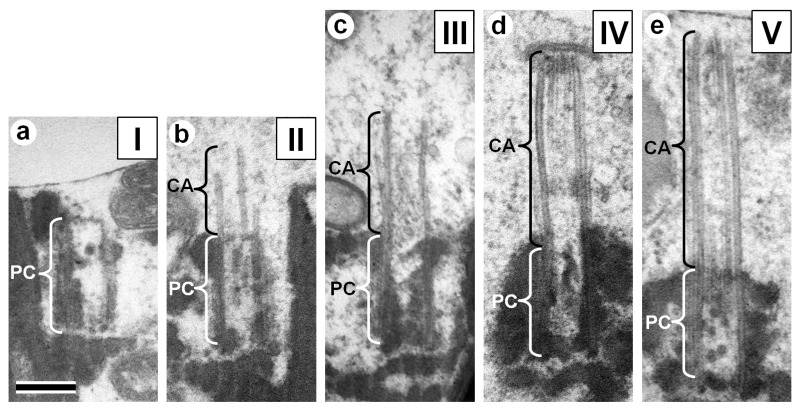
TEM images illustrating the different length classes of the PC+CA complexes in human spermatozoa. (**a**) Class I: 300–499 nm. (**b**) Class II: 500–699 nm. (**c**) Class III: 700–899 nm. (**d**) Class IV: 900–1099 nm. (**e**) Class V: 1100–1299 nm. Photos (**a**) and (**b**) from healthy donor spermatozoa, photos **(c**–**e**) from IMS patient spermatozoa. CA: centriolar adjunct; PC: proximal centriole. Scale bar: 200 nm.

**Figure 4 cells-08-00180-f004:**
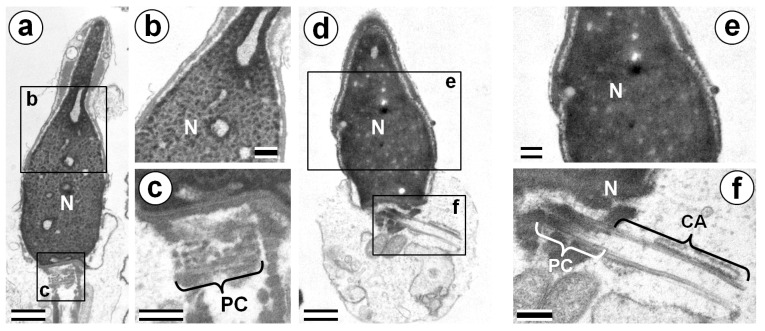
TEM analysis of chromatin of P1 spermatozoa nuclei. (**a**–**c**) Cells with incomplete chromatin condensation and short PC+CA complexes. (**d**–**f**) Cells with complete chromatin condensation and long PC+CA complexes. CA: centriolar adjunct; N: nucleus; and PC: proximal centriole. Scale bar: 500 nm in (**a**), (**c**), (**d**), and (**f**); and 200 nm in (**b**) and (**e**).

**Table 1 cells-08-00180-t001:** Comparative analysis of the length of the PC+CA complex and the occurrence of spermatozoa with incomplete chromatin condensation in fertile donors (D1–D5) and IMS patients (P1, P2).

Donors (D1–D5) and Patients (P1, P2)	D1	D2	D3	D4	D5	P1	P2
Length of complex PC+CA (nm)	560 ± 150	555 ± 141	569 ± 162	543 ± 111	560 ± 132	1059 ± 100	769 ± 160
% of spermatozoa without centriolar adjunct (I class)	43.3	43.3	33.3	43.3	36.7	3.3	6.7
% of spermatozoa population with complete condensation of chromatin	83.6 (n = 535)	81.2 (n = 521)	72.6 (n = 594)	85.1 (n = 510)	88.8 (n = 500)	39.9 (n = 501)	88.8 (n = 525)
% of “normal spermatozoa” by two parameters: centriolar adjunct disassembly and chromatin condensation	36.2	35.2	24.1	36.8	32.6	1.3	5.9

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
