# Peer review of "The Centriolar Adjunct–Appearance and Disassembly in Spermiogenesis and the Potential Impact on Fertility"

_cells, 2019, doi:10.3390/cells8020180_

Round 1
Reviewer 1 Report
The paper addresses an important subject in cell biology of spermatids – the centriolar adjunct. Overall, I enjoyed reading the paper, and I believe it constitutes a significant contribution because it presents a novel phenotype that is associated with unexplained infertility. The article should be published after adding a discussion about the points I mentioned in major concerns as well as addressing the minor concerns.
Major concerns
- “The centriolar adjunct demonstrates dynamic changes in size, decreasing during spermatozoa maturation, and disappears partially or completely in mature spermatozoa [13, 17, 18].” This statement is not accurate as it was claimed in the past that the centriolar adjunct remains in human spermatozoon [see ZAMBONI, L. & STEFANINI, M. 1971. The fine structure of the neck of mammalian spermatozoa. Anat Rec, 169, 155-72. They write "It is, therefore, not unreasonable to consider the frequent persistence of the centriolar adjunct in human ejaculated spermatozoa, also reported by De Kretser ('69), as a character of relative immaturity, just as the primitive type of aggregation of the chromatin and the persistence of a voluminous cytoplasmic droplet (Zamboni et al., '71)."]
-“ This study strongly suggests, for the first time, that disassembly of the centriolar adjunct is required for the functional maturity of human spermatozoa.” This is an overstatement since the authors did exclude other possibilities. They must be qualified and add that “however, other mechanisms are possible, and the centriolar adjunct may be a pleiotropic phenotype.”
- “Thus, we can conclude that disassembly of the centriolar adjunct is required for the functional maturity of man spermatozoa, and it does not depend on nucleus chromatin condensation.” However, in table 1, P#1, which cannot have kids has 60.1% (n=501) of spermatozoa population with incomplete condensation of chromatin, which is higher than all the other samples that can have kids. Can this be the reason P#1 have no kid? – this needs to be discussed
Minor concerns
- If it is known, it would be useful to add if the partners of Patient# 1 and #2 had any phenotypes that may contribute to infertility.
-“The wall of the centriolar adjunct consists of electron-dense material with nine complete MTs “A” and 18 incomplete MTs “B” and “C” immersed in it.” Please provide a high magnification picture that shows this.
-“The function of the centriolar adjunct is still unknown.” Please add and site that the centriolar adjunct was proposed to play a role in determining the plane of the nucleus flattening.
- “Indeed, boars with 100% fertility have proximal centrioles with a completely disassembled centriolar adjunct”. What is “100% fertility” and how is it determined? What are the references for that?
“Based on the structure of the CA we propose that this structure might be partial analogous to another centriole derivate, the primary cilium.” Please add that the CA also resembles a cytosolic cilium because it is not surrounded by a ciliary membrane (see AVIDOR-REISS, T. & LEROUX, M. R. 2015. Shared and Distinct Mechanisms of Compartmentalized and Cytosolic Ciliogenesis. Curr Biol, 25, R1143-50).
-“both primary cilia and CA are growing from the mother centriole [21]”. This point is controversial as the distal centriole that forms the flagellum is thought by most scientists to be the mother centriole. Please point out this controversy in the paper.
Author Response
The paper addresses an important subject in cell biology of spermatids – the centriolar adjunct. Overall, I enjoyed reading the paper, and I believe it constitutes a significant contribution because it presents a novel phenotype that is associated with unexplained infertility. The article should be published after adding a discussion about the points I mentioned in major concerns as well as addressing the minor concerns.
Major concerns
- “The centriolar adjunct demonstrates dynamic changes in size, decreasing during spermatozoa maturation, and disappears partially or completely in mature spermatozoa [13, 17, 18].” This statement is not accurate as it was claimed in the past that the centriolar adjunct remains in human spermatozoon [see ZAMBONI, L. & STEFANINI, M. 1971. The fine structure of the neck of mammalian spermatozoa. Anat Rec, 169, 155-72. They write "It is, therefore, not unreasonable to consider the frequent persistence of the centriolar adjunct in human ejaculated spermatozoa, also reported by De Kretser ('69), as a character of relative immaturity, just as the primitive type of aggregation of the chromatin and the persistence of a voluminous cytoplasmic droplet (Zamboni et al., '71)."]
Human spermatozoa showing persistence of the centriolar adjunct (Fawcett and Phillips, '69). This structure, which is shown in longitudinal section in figures 8 and 9, and in cross section in figures 10 and 11, resembles a centriole in its organization. It differs from the latter mainly for presence of doublets and singlets in association with triplets, open outermost subfibers, and presence of a circular layer of material of difficult resolution on the interior of the fibers.
Authors‘s response (AR): We agreed with reviewer’s comment and have revised the text (lines 72-79, marked in blue). The article of Zamboni, L.; Stefanini, 1971 was added in the list of references. This work is in line with our observations and confirmed that human spermatozoa differ in the degree of disassembly of a centriolar adjunct from not only boar spermatozoa, but also from sperm cells of rabbits and other primates.
-“ This study strongly suggests, for the first time, that disassembly of the centriolar adjunct is required for the functional maturity of human spermatozoa.” This is an overstatement since the authors did exclude other possibilities. They must be qualified and add that “however, other mechanisms are possible, and the centriolar adjunct may be a pleiotropic phenotype.”
AR: We agreed with this remark and made the statement less categorical (lines 93-94, marked in blue).
- “Thus, we can conclude that disassembly of the centriolar adjunct is required for the functional maturity of man spermatozoa, and it does not depend on nucleus chromatin condensation.” However, in table 1, P#1, which cannot have kids has 60.1% (n=501) of spermatozoa population with incomplete condensation of chromatin, which is higher than all the other samples that can have kids. Can this be the reason P#1 have no kid? – this needs to be discussed.
AR: We revealed all four possible variants of combinations: spermatozoa with condensed chromatin and long adjunct, spermatozoa with condensed chromatin and disassembled (short) adjunct, spermatozoa with incomplete condensed chromatin and long adjunct, and finally, spermatozoa with incomplete condensed chromatin and disassembled adjunct (Fig. 4). Therefore, we assume that incomplete chromatin condensation and incomplete disassembly of the centriolar adjunct are two independent of each other sperm maturation disorders.
For a clearer understanding of the combined effect of the two factors studied, we have modified Table 1 (lines 312-315 and in the text lines 336-343) by i) adding a line “proportion of spermatozoa with no adjunct (class 1)”; ii) replacing the percentage of cells with incompletely condensed chromatin by the percentage of cells with fully condensed chromatin; and ii) adding a line “percentage of "normal sperm". As can be seen from the table, donors had 4-28 times more normal spermatozoids than patients (relative to only these parameters); it is possible that other factors may reduce a part of normal sperm patients to zero). Directly answering your question, we can say that the presence of a high rate of incomplete chromatin condensation makes the first patient “more sterile” than the second. The second patient has theoretically a higher chance to conceive a child (for example by multiple attempts using ART).
Minor concerns
- If it is known, it would be useful to add if the partners of Patient# 1 and #2 had any phenotypes that may contribute to infertility.
AR: Patient 1's partner had no phenotypes that could contribute to women infertility. We added this phrase to the text (lines 104-105).
The partner of the second patient gave birth to a healthy girl at the first use of a healthy donor sperm for the fertilization. This information is given in the Materials and Methods (lines 109-111).
-“The wall of the centriolar adjunct consists of electron-dense material with nine complete MTs “A” and 18 incomplete MTs “B” and “C” immersed in it.” Please provide a high magnification picture that shows this.
AR: In the new version of Figure 1 we removed the inserts; the cross-sections of the centriole, centriolar adjunct, and flagellum are shown in separate, significantly enlarged photographs (New Fig. 1 e-g).
-“The function of the centriolar adjunct is still unknown.” Please add and site that the centriolar adjunct was proposed to play a role in determining the plane of the nucleus flattening.
AR: We’ve added this hypothesis and correspondent results of our test of this in the text (lines 180-185).
It was suggested that a centriolar adjunct can play a role in determining the plane of the nucleus flattening (mechanism of this “regulation” is unclear) [19]. Like the spermatids nucleus of a goat, porcine spermatids had flattened ovoid nuclei. We can confirm the data obtained in work [19] that the position of the centriolar adjunct was not accidental, together with the flagellum it formed a plane parallel to the plane of the spermatid nucleus flattening (N=10).
- “Indeed, boars with 100% fertility have proximal centrioles with a completely disassembled centriolar adjunct”. What is “100% fertility” and how is it determined? What are the references for that?
AR: We revised the text (line 259-262). Indeed, we use the sperm of normal fertile boar (100% success of inseminations in experimental farm, that was revealed by the birth of normal piglets by all the inseminated sows).
All the spermatozoids of fertile boar had proximal centrioles with a completely disassembled centriolar adjunct (Fig. 1d, h-j). We can propose that fertilization capacity is proportional to the ratio of spermatozoa with a fully or almost completely disassembled centriolar adjunct (PC+CA complexes length class I).
- “Based on the structure of the CA we propose that this structure might be partial analogous to another centriole derivate, the primary cilium.” Please add that the CA also resembles a cytosolic cilium because it is not surrounded by a ciliary membrane (see AVIDOR-REISS, T. & LEROUX, M. R. 2015. Shared and Distinct Mechanisms of Compartmentalized and Cytosolic Ciliogenesis. Curr Biol, 25, R1143-50).
AR: We agree with this reviewer’s comment, and added the appropriate modifications (lines 272-273, marked in blue). The reference (Avidor-Reiss, Leroux, 2015) was added.
… in particular, the primary cytosolic cilia, which is not yet surrounded by a membrane at this early stage of formation [22].
- “both primary cilia and CA are growing from the mother centriole [21]”. This point is controversial as the distal centriole that forms the flagellum is thought by most scientists to be the mother centriole. Please point out this controversy in the paper.
AR: This controversy was pointed in the revised version as suggested (line 276-279).
It thought sometimes, that distal centriole which forms the flagellum is mother centriole (like mother centriole which forms primary cilium in somatic cells); however, our data obtained on early porcine spermatids [23], showed that this is the daughter centriole that forms a flagellum.
Indeed, during the discussion on our first results at the conference “Centrosome & Spidle Pole Bodies " in Barcelona in 2011, the opinions of the experts (Michel Bornens, Greenfield Sluder, Erich Nigg, Jordan Raff, Alexey Khodjakov, Pierre Gönczy and others) were divided. However, the majority of our colleagues considered the motile flagellum a spermatozoon derivative of the daughter centriole, as well as the centriolar adjunct is derivative of mother centriole. Later we specifically investigated this issue in pig spermatids at various stages of spermiogenesis (Alieva et al., 2018, ref 23). The principal figure from this publication is available at this link:
https://www.researchgate.net/publication/327646211_Chapter_5_A_QUESTION_OF_FLAGELLA_ORIGIN_FOR_SPERMATIDS_-_MOTHER_OR_DAUGHTER_CENTRIOLE
From presented TEM photos of spermatids on Figure 10 (during later stages of spermiogenesis, the centrioles shift relative to each other, and at the earlier stage flagella is absent) it is obvious that the flagellum of the cilia grows from the daughter centriole (proximal end of daughter centriole is associated with the lateral surface of mother centriole). We also see in the photograph anlage of capitulum associated with the proximal (mother) centriole and anlage of segmented columns, associated with the distal (daughter) centriole. Consequently, the distal centriole giving rise to flagella is a daughter centriole, and the proximal centriole giving rise to the centriolar adjunct is the mother centriole.
I want to thank you for a detailed analysis of our work and constructive criticism, which, we dare to hope, has made our article more understandable and useful for readers.
Reviewer 2 Report
The manuscript entitled “The centriolar adjunct – forgotten cousin of the primary cilium. Its appearance and resorbtion in spermiogenesis and the potential impact on fertility” from
R.E. Uzbekov and co-authors provides the TEM analyses of the centriolar adjunct in porcine spermatids and in human spermatozoa obtained from fertile donors (control) and IMS patients. Based on the purely microscopic analyses Authors speculate that:
1. disassembly of the centriolar adjunct is required for the functional maturity of human spermatozoa
2. incomplete centriole adjunct disassembly could be a reason for some development dysfunctions.
3. The length of the centriolar adjunct can be potentially used in the diagnostics of IMS patients.
4. CA is partially analogous to the primary cilium.
Although the manuscript is clearly written and provides interesting TEM analyses, I have some issues. The existence of CA was described before, so it is not a complete novelty. What is unknown, is the function of this structure. The submitted manuscript does not provide experimental data pointing to the function of the CA but rather a description of the changes of CA during the spermatozoa maturation and in infertile individuals. Based on TEM data Authors speculate about a role of CA, and it possible analogy to the primary cilium and possible role in diagnostics. The latter, in my opinion, are far-fetched. Authors analyzed sperm cells from only two IMS patients and obtained data concerning PC+CA length varied significantly between them. I regret to say, but the number of analyzed individuals is too low to draw conclusions, especially with such variability.
I also think that as it is now, the manuscript is more suitable for a journal like Reproduction rather than journal focused on general biology.
Additional remarks:
Title
A statement in the title: “ The centriolar adjunct – forgotten cousin of the primary cilium” is rather far-fetched and might be misleading. This manuscript does not provide any experimental data indicating that a centriolar adjunct and primary cilia are analogous organelles.
If I am not mistaken it should be resorption (with “p”)
Abstract
“The average length of the “proximal centriole + centriolar adjunct” complex measured in sections, where it had parallel orientation in the section plane, was significantly longer in the spermatozoa of IMS patients than in the spermatozoa of healthy donors.”
Please change: “The average length….. was significantly longer…”
Introduction
lines 43-54
This fragment should be transferred to the discussion section. The content of the manuscript does not directly relate to the assisted reproductive technology but Authors hypothesis (that the centriolar adjunct length can be potentially used in the diagnostics) can be discussed.
Lines 55-56:
“Several important morphological changes occur in spermatid maturation during last part of spermiogenesis during transformation into mature sperm cells.” – please rephrase.
Line 93: “The aim of this study was to investigate the reason for infertility…” suggested a change: to investigate a cause …
Results
Fig. 1A – insets S1 and S2 should be bigger – this is a description of CA, so the structure of CA should be clearly visible.
Fig. 1B – the boundary between the proximal centriole and CA is well defined. What is the structure of this boundary? Does it resemble the structure of the transition zone?
Fig. 1C – a dense material is visible at the level of the boundary between a proximal centriole and CA. Is this a collar-like structure? It is still visible in Fig. 1e
Fig. 2E – “…different number of asterisks means significant differences between the data groups”. Why P1 has two and P2, has three asterisks.
Fig. 2I – the 3D structure is not apparent on the presented reconstructed model of PC+CA in IMS patient. Please replace the image (maybe change the angle), please add reconstruction of PC+CA in a healthy individual.
Fig. 2D and K – a structure of the proximal centriole looks different in a healthy individual and IMS patient. Please comment.
Fig. 2A and G; Fig. 3B and D (and Supplementary Figs S1 and S2) – a proximal centriole in IMS patient seems to be longer. How often is it impossible to define a boundary between PC and CA? In case such boundary can be defined, is indeed PC longer in IMS patient?
Lines 221-224: “Exploring the spermatozoa of healthy donors (as controls) and IMS patients, we found that, in some cells, the proximal centrioles were associated with structures similar to centriolar adjuncts, which we describe in porcine spermatids (Fig. 1 and 2). “– Please provide the % and N (number of all cells analyzed) of spermatozoa with CA (for healthy individuals and IMS patients).
Lines 224-226: “there was no electron-dense material in the walls and no discs in the internal volume of human spermatozoa centriolar adjuncts.” – If I am not mistaken this sentence describes a CA in mature human spermatozoa. CA with discs and electron dense material was observed in pig spermatids. Is this correct?
Lines 230-231: “In some cases, the border between the centriolar adjunct (CA) and the proximal centriole (PC) was not sufficiently obvious” – please provide % and N of the analyzed cells
Lines 267-268: “Fertility is proportional to the ratio of spermatozoa with a fully or almost completely disassembled centriolar adjunct (PC+CA complexes length class I).” – What is the base for this statement?
Lines 269-270: “Indeed, boars with 100% fertility have proximal centrioles with a completely disassembled centriolar adjunct” – add REF. Do you mean male pig? Are there data showing that boars/ male pigs with CA has lower fertility? REF? How the fertility of male pigs was estimated?
Lines 278-282: “Based on the structure of the CA we propose that this structure might be partial analogous to another centriole derivate, the primary cilium.” Authors claim that based on: “CA does not have central MTs” and that “both primary cilia and CA are growing from the mother centriole”.
However, as Authors stated above, opposite to the ciliary axoneme:” The wall of the centriolar adjunct consists of electron-dense material with 9 complete MTs “A” and 18 incomplete MTs “B” and “C” immersed in it.” Presence of triplets contradicts the above hypothesis. In a very general way all structures formed on the base of centriole / basal body can be analog, not only primary cilium and CA.
The next Authors hypothesis (“These data allowed us to hypothesize that the CA may also be a regulator of first mitotic divisions, zygote cleavages”) is based on the first, yet unproven hypothesis that CA is an analogue of a primary cilium.
Lines 290-293: “Malfunction of CA disassembly during spermiogenesis might indicate anomalies in regulatory kinase systems, and CA might serve as a kind of” brake” on a faulty embryonic process.” – Authors have no basis for this claim as kinase system was not investigated.
Lines 342-343:” Thus, we can conclude that disassembly of the centriolar adjunct is required for the functional maturity of human spermatozoa,..”. However, as Authors indicated above: “…spermatozoa with condensed chromatin and disassembled (short) adjunct..” were present in patients (lines 323-328, statistics is missing). So there are other factors causing infertility.
Author Response
Comments and Suggestions for Authors
The manuscript entitled “The centriolar adjunct – forgotten cousin of the primary cilium. Its appearance and resorbtion in spermiogenesis and the potential impact on fertility” from
R.E. Uzbekov and co-authors provides the TEM analyses of the centriolar adjunct in porcine spermatids and in human spermatozoa obtained from fertile donors (control) and IMS patients. Based on the purely microscopic analyses Authors speculate that:
1. disassembly of the centriolar adjunct is required for the functional maturity of human spermatozoa
2. incomplete centriole adjunct disassembly could be a reason for some development dysfunctions.
3. The length of the centriolar adjunct can be potentially used in the diagnostics of IMS patients.
4. CA is partially analogous to the primary cilium.
Although the manuscript is clearly written and provides interesting TEM analyses, I have some issues. The existence of CA was described before, so it is not a complete novelty. What is unknown, is the function of this structure. The submitted manuscript does not provide experimental data pointing to the function of the CA but rather a description of the changes of CA during the spermatozoa maturation and in infertile individuals. Based on TEM data Authors speculate about a role of CA, and it possible analogy to the primary cilium and possible role in diagnostics. The latter, in my opinion, are far-fetched. Authors analyzed sperm cells from only two IMS patients and obtained data concerning PC+CA length varied significantly between them. I regret to say, but the number of analyzed individuals is too low to draw conclusions, especially with such variability.
I also think that as it is now, the manuscript is more suitable for a journal like Reproduction rather than journal focused on general biology.
Author responce (AR): Although the percentage of men with fertility problems is quite high, most of the causes of abnormalities are diagnosed already at the level of study in a light microscope. Ultrastructural diagnosis of spermatozoa is quite expensive and is still not widely distributed. But even when patients' spermatozoa are examined with an electron microscope, in the overwhelming majority of cases, this study is limited to single sections to detect chromatin condensation abnormalities. In this case, research on serial sections of the centriolar region is not carried out. A broad ultrastructural study of spermatozoa has been performed at two Reproduction centers in Moscow and Tours. The first patient (patient 2 in this work) was discovered by us at the Moscow Center in 2009. Despite the fact that his percentage of cells with incomplete chromatin condensation was on the same level as donors, the development of all zygotes stopped at an early stage. We studied of the centriole structure and reported the results of our work at a conference in Barcelona (Alieva et al., 2011). We had to wait for the second similar patient for more than 5 years after this (patient 1 in this paper from Tours Reproduction center). Initially, an ultrastructural study of his sperm was not conducted. It turned out that this patient had both types of developmental disorders — an increased proportion of spermatozoa with incomplete chromatin condensation and impaired disassembly of the centriolar adjunct.
Thus, it is not possible to obtain a larger number of patients for detailed research for two objective reasons: 1 - there are few such patients; 2- most of those who have similar problems do not try to establish the true cause of male sterility (they do not pay ultrastructural study), but immediately use sperm donor services.
It seems to us that not only the data from patients with IMS, but also the comparison of spermatozoa of healthy fertile donors and the fertile boar given in this article can be useful for understanding the role of a centriolar adjunct. Although there is conflicting evidence in the literature about disassembling a centriolar adjunct in human spermatozoa (the diagrams or photos show a human proximal centriole both with [3, 18] or without adjunct [2, 13], this process never was quantified even for healthy fertile donors.
Regarding the comparison of the centriolar adjunct and primary cilia, we believe that this is one of the main observations obtained as a result of our work in recent years. Two systems of terminology (proximal and distal centriole in sperm) and (mother and daughter centriole in somatic cells) existed in parallel for many years. Now it is possible to correlate these two terminologies.
For the first time, we received direct evidence that the sperm flagella grow from a daughter, and not from a mother centriole (Alieva et al. 2018), and answered one of the key questions that worried cell biologists. This is what gave us reason to call the centriolar adjunct cousin of primary cilia. It is, of course, not a twin or even not a brother, but his (her) cousin in our opinion. This allows us to extrapolate recent results on the role of primary cilia in the regulation of mitosis to the centriolar adjunct.
In our opinion, these results are important and interesting not only for specialists in reproduction biology, but also for a wide range of cellular biologists, which gave us reason to submit our article in the journal Cells, which has a wider readership than specialized reproductive journals.
Additional remarks:
Title
A statement in the title: “The centriolar adjunct – forgotten cousin of the primary cilium” is rather far-fetched and might be misleading. This manuscript does not provide any experimental data indicating that a centriolar adjunct and primary cilia are analogous organelles.
AR : What types of experiments can confirm or deny this? In this case, we proceed from the structure (absence of central MTs), the absence of motor function and connection with the mother centriole. The last principal statement was confirmed by us during the study of spermatids at various stages of development in our previous work (Alieva et al., 2018). The principal figure from this publication is available at this link:
https://www.researchgate.net/publication/327646211_Chapter_5_A_QUESTION_OF_FLAGELLA_ORIGIN_FOR_SPERMATIDS_-_MOTHER_OR_DAUGHTER_CENTRIOLE
From presented TEM photos of spermatids on Figure 10 (during later stages of spermiogenesis, the centrioles shift relative to each other, and at the earlier stage flagella is absent) it is obvious that the flagellum of the cilia grows from the daughter centriole (proximal end of daughter centriole is associated with the lateral surface of mother centriole). We also see in the photograph anlage of capitulum associated with the proximal (mother) centriole and anlage of segmented columns, associated with the distal (daughter) centriole. Consequently, the distal centriole giving rise to flagella is a daughter centriole, and the proximal centriole giving rise to the centriolar adjunct is the mother centriole.
If I am not mistaken it should be resorption (with “p”)
AR : “resorption” is reabsorption
“resorbtion” is progressive disassembly, disappearance
We thus used the correct definition of this process.
Abstract
“The average length of the “proximal centriole + centriolar adjunct” complex measured in sections, where it had parallel orientation in the section plane, was significantly longer in the spermatozoa of IMS patients than in the spermatozoa of healthy donors.”
Please change: “The average length….. was significantly longer…”
AR: If we understand correctly, you suggest to change the word "longer" to “greater”.
We’ve done it. (line 34)
Introduction
lines 43-54
This fragment should be transferred to the discussion section. The content of the manuscript does not directly relate to the assisted reproductive technology but Authors hypothesis (that the centriolar adjunct length can be potentially used in the diagnostics) can be discussed.
AR: We agree with your recommendation and transfer this section to the end of the paper (lines 346-361). Additionally, we wrote small conclusion.
This paper gives an explanation of one of the reasons for stopping the development of a zygote after fertilization in vitro and transfers this case from the category of «idiopathic» to the category of diagnosable disorders of sperm development. (lines 361-364)
Lines 55-56:
“Several important morphological changes occur in spermatid maturation during last part of spermiogenesis during transformation into mature sperm cells.” – please rephrase.
“Several important morphological changes occur in spermatid maturation during last part of spermiogenesis during transformation their into mature sperm cells.”
AR: We added “their” in the sentence. (line 47)
Line 93: “The aim of this study was to investigate the reason for infertility…” suggested a change: to investigate a cause …
AR: We agree with this comment and made a change in the text. (line 89)
Results
Fig. 1A – insets S1 and S2 should be bigger – this is a description of CA, so the structure of CA should be clearly visible.
AR : We agree with this remark and presented the inserts in the form of individual photos in modified figure 1. (lines 189-199)
Fig. 1B – the boundary between the proximal centriole and CA is well defined. What is the structure of this boundary? Does it resemble the structure of the transition zone?
AR: In Figure 1B, in the region of the boundary between the centriole and centriolar adjunct, the tangentially cut part of the first inner ring of the centriolar adjunct is visible (see the previous section in Photo 1a). In the lumen of the centriolar cylinder, irregularly dense granules were sometimes found (see Figs. 2a, 2d, 2g, 2j, 3a, 3e, 4b), but no additional structures such as the terminal plate were found.
Fig. 1C – a dense material is visible at the level of the boundary between a proximal centriole and CA. Is this a collar-like structure? It is still visible in Fig. 1e
AR: It seems that «Сollar-like structure» is special terminus from another area. https://www.ncbi.nlm.nih.gov/pubmed/2417845
In this case, we think that it is part of striated columns that are adjacent to the proximal centriole from the outside.
Fig. 2E – “…different number of asterisks means significant differences between the data groups”. Why P1 has two and P2, has three asterisks.
AR: A different number of stars between P1 and P2 shows that they are not only statistically different from donors, but also from each other. Centriolar adjunct in P1 patient spermatozoa usually longer then in spermatozoa of P1 patient.
Fig. 2I – the 3D structure is not apparent on the presented reconstructed model of PC+CA in IMS patient. Please replace the image (maybe change the angle), please add reconstruction of PC+CA in a healthy individual.
AR: The aim of the 3D reconstruction was to show in a volumetric form the continuity of the MT of the proximal centriole and centriolar adjunct. The selected angle, moreover, shows optimally the separation of triplets and doublets into separate MTs at the end of the centriolar adjunct.
The reconstruction of the centriole of the sperm cell of the fertile donor was added by us as new Fig. S1 in two orientations with serial TEM cross sections.
Fig. 2D and K – a structure of the proximal centriole looks different in a healthy individual and IMS patient. Please comment.
AR: This small difference (angle of divergence of triplets) is caused by a difference in the level of the section. As is known (Vorobjev, Chentsov, 1982), the triplets at the proximal end of the centrioles are more “unfolded”. In cat spermatozoa this small difference was shown in the paper Sato and Oura, 1984 (Fig. 5).
Fig. 2A and G; Fig. 3B and D (and Supplementary Figs S1 and S2) – a proximal centriole in IMS patient seems to be longer. How often is it impossible to define a boundary between PC and CA? In case such boundary can be defined, is indeed PC longer in IMS patient?
AR: Unlike in the spermatids, the boundary between the centriolar adjunct and the proximal centriole cannot be accurately determined. That is why we measured the total length of the entire complex. As can be seen from the data on the length of centrioles in boar spermatozoa (Fig. 1d), the length of centrioles is not a strictly deterministic value, but has slight variations.
Lines 221-224: “Exploring the spermatozoa of healthy donors (as controls) and IMS patients, we found that, in some cells, the proximal centrioles were associated with structures similar to centriolar adjuncts, which we describe in porcine spermatids (Fig. 1 and 2). “– Please provide the % and N (number of all cells analyzed) of spermatozoa with CA (for healthy individuals and IMS patients).
AR: For each patient and donor, we have examined 2000–3000 spermatozoa. The percentage of centrioles with adjunct in patients was more than 90%, in donors about 50%. However, the data obtained on oblique sections do not make any sense, since they do not allow a quantitative analysis of the length of adjuncts. In addition, the oblique oriented centriole and adjunct lies on several successive sections, and it is impossible to determine at what distance in the volume of the last section the structure ends. Thus, situations "no adjunct" and "short adjunct" cannot be differ on oblique sections.
The sections parallel to the long axis of the complex make up only about 1 percent of all possible orientations in the object space. Thus, in order to find 30 suitable spermatozoa for analysis, we had to examine 2-3 thousand (two years of work!). This is exactly the case when§ in our opinion, a smaller but reliable sample provides much more information than a much larger, but not sufficiently reliable.
Lines 224-226: “there was no electron-dense material in the walls and no discs in the internal volume of human spermatozoa centriolar adjuncts.” – If I am not mistaken this sentence describes a CA in mature human spermatozoa. CA with discs and electron dense material was observed in pig spermatids. Is this correct?
AR: You understood correctly. We could not examine human testicles (ethics requirements) therefore we’ve compared the structure of residual centriolar adjuncts in sperm of humans with spermatids in pigs. As far as we know, the centriolar adjunct has been specially studied in human spermatids only in one work De Kretser, 1969. Structure of centriolar adjunct in human spermatids in this work was practically identical to our photos from porcine spermatids.
Don W Fawcett in the second edition (1981) of the book "The Cell" (Chapter 12: Centrioles) present centriolar adjunct in chinchilla spermatids (Fig. 314), which was less similar (internal lumen structure had not visible periodic structures) to a human one than porcine centriolar adjunct. On contrary, in the article of Kojima (1994) internal centriolar adjunct structure in goat (Fig. 3) was practical identical to porcine centriolar adjunct in our paper and human centriolar adjunct in De Kretser (1969, Fig. 11) paper.
Lines 230-231: “In some cases, the border between the centriolar adjunct (CA) and the proximal centriole (PC) was not sufficiently obvious” – please provide % and N of the analyzed cells
AR:We change phrase to: “The border between the centriolar adjunct (CA) and the proximal centriole (PC) was not obvious” (lines 210-211)
Lines 267-268: “Fertility is proportional to the ratio of spermatozoa with a fully or almost completely disassembled centriolar adjunct (PC+CA complexes length class I).” – What is the base for this statement?
AR: We rewrite this paragraph:
All the spermatozoids of fertile boar had proximal centrioles with a completely disassembled centriolar adjunct (Fig. 1d, h-j). We can propose that fertilization capacity is proportional to the ratio of spermatozoa with a fully or almost completely disassembled centriolar adjunct (PC+CA complexes length class I). (lines 260-263)
Lines 269-270: “Indeed, boars with 100% fertility have proximal centrioles with a completely disassembled centriolar adjunct” – add REF. Do you mean male pig? Are there data showing that boars/ male pigs with CA has lower fertility? REF? How the fertility of male pigs was estimated?
AR: All the spermatozoids of fertile boar had proximal centrioles with a completely disassembled centriolar adjunct (Fig. 1d, h-j). We can have proposed that fertilization capacity is proportional to the ratio of spermatozoa with a fully or almost completely disassembled centriolar adjunct (PC+CA complexes length class I).
Lines 278-282: “Based on the structure of the CA we propose that this structure might be partial analogous to another centriole derivate, the primary cilium.” Authors claim that based on: “CA does not have central MTs” and that “both primary cilia and CA are growing from the mother centriole”.
However, as Authors stated above, opposite to the ciliary axoneme:” The wall of the centriolar adjunct consists of electron-dense material with 9 complete MTs “A” and 18 incomplete MTs “B” and “C” immersed in it.” Presence of triplets contradicts the above hypothesis. In a very general way all structures formed on the base of centriole / basal body can be analog, not only primary cilium and CA.
The next Authors hypothesis (“These data allowed us to hypothesize that the CA may also be a regulator of first mitotic divisions, zygote cleavages”) is based on the first, yet unproven hypothesis that CA is an analogue of a primary cilium.
AR: We do not say that these structures are identical, like brothers or even twins; we say they look like cousins. This is really our hypothesis, which is based on growth from the mother centriole, the absence of central MTs and the absence of obvious motor function. Since we have a delayed division of zygotes associated with an unparsed centriolar adjunct, we draw a parallel with the delayed mitotic division of somatic cells when the primary cilia are not disassembled. The two hypotheses, when combined, give a logically substantiated confirmation of possible affinity, not only by origin (from the mother centriole), but also functionally by the primary cilia and the centriolar adjunct.
Lines 290-293: “Malfunction of CA disassembly during spermiogenesis might indicate anomalies in regulatory kinase systems, and CA might serve as a kind of” brake” on a faulty embryonic process.” – Authors have no basis for this claim as kinase system was not investigated.
AR: This is only a hypothesis based on analogy with the data obtained for the primary cilia. Why not?
Lines 342-343:” Thus, we can conclude that disassembly of the centriolar adjunct is required for the functional maturity of human spermatozoa...”. However, as Authors indicated above: “…spermatozoa with condensed chromatin and disassembled (short) adjunct.” were present in patients (lines 323-328, statistics is missing). So there are other factors causing infertility.
AR: No doubt, you are right. Other factors and among them incomplete chromatin condensation have their effect on fertility. We have included in the Table 1 additional columns: "% of spermatozoa without centriolar adjunct (I class)" and “% of “normal spermatozoa” from two parameters: centriolar adjunct disassembly and chromatin condensation” and evaluated the decline in fertility in the aggregate of the two parameters studied for two patients and five donors. These two parameters, of course, do not completely cover all the factors affecting fertility, but they allow to estimate it approximately.
Abnormality of the various components of the flagellar axoneme occasionally may be noted in the spermatozoa of normal individuals (Zamboni, 1987, Pryor et al. 1981, Hunter and Deretzer 1986). The same can be said about other defects of sperm. Final fertility (or sterility) will depend on the totality of all disorders. For patients studied in our work only by two parameters, fertility decreased by 4-28 times in compare with fertile donors (who are not absolutely fertile too, like we see!). This led to the practical impossibility of developing normal embryos even when fertilized in vitro by pre-selected spermatozoids by morphological parameters.
I want to thank you for a detailed analysis of our work and constructive criticism, which, we dare to hope, has made our article more understandable and useful for readers.
Round 2
Reviewer 1 Report
Add scale bar to figure 1a-c
Author Response
Add scale bar to figure 1a-c
AR 2: Thanks, it's done.
We added scale bar to figure 1c (common for to figure 1a-c).
Moderate English changes required
AR 2: First version of the paper was checked by Proof-Reading-Service.com Ltd (Devonshire, company registration number 8391405) and final version after second revision was corrected by Dr. Michael Jennings, who is cell biologist and native English speaker (green font color in the text).
I want to thank you for a detailed analysis of our work and constructive criticism, which, we dare to hope, has made our article more understandable and useful for readers.
Reviewer 2 Report
The manuscript was improved. Most of the comments was addressed, presented experimental data are scientifically sound. The specific comments are indicated in red in the attached file.
Author's Notes
Comments and Suggestions for Authors
The manuscript entitled “The centriolar adjunct – forgotten cousin of the primary cilium. Its appearance and resorbtion in spermiogenesis and the potential impact on fertility” from
R.E. Uzbekov and co-authors provides the TEM analyses of the centriolar adjunct in porcine spermatids and in human spermatozoa obtained from fertile donors (control) and IMS patients. Based on the purely microscopic analyses Authors speculate that:
1. disassembly of the centriolar adjunct is required for the functional maturity of human spermatozoa
2. incomplete centriole adjunct disassembly could be a reason for some development dysfunctions.
3. The length of the centriolar adjunct can be potentially used in the diagnostics of IMS patients.
4. CA is partially analogous to the primary cilium.
Although the manuscript is clearly written and provides interesting TEM analyses, I have some issues. The existence of CA was described before, so it is not a complete novelty. What is unknown, is the function of this structure. The submitted manuscript does not provide experimental data pointing to the function of the CA but rather a description of the changes of CA during the spermatozoa maturation and in infertile individuals. Based on TEM data Authors speculate about a role of CA, and it possible analogy to the primary cilium and possible role in diagnostics. The latter, in my opinion, are far-fetched. Authors analyzed sperm cells from only two IMS patients and obtained data concerning PC+CA length varied significantly between them. I regret to say, but the number of analyzed individuals is too low to draw conclusions, especially with such variability.
I also think that as it is now, the manuscript is more suitable for a journal like Reproduction rather than journal focused on general biology.
Author responce (AR): Although the percentage of men with fertility problems is quite high, most of the causes of abnormalities are diagnosed already at the level of study in a light microscope. Ultrastructural diagnosis of spermatozoa is quite expensive and is still not widely distributed. But even when patients' spermatozoa are examined with an electron microscope, in the overwhelming majority of cases, this study is limited to single sections to detect chromatin condensation abnormalities. In this case, research on serial sections of the centriolar region is not carried out. A broad ultrastructural study of spermatozoa has been performed at two Reproduction centers in Moscow and Tours. The first patient (patient 2 in this work) was discovered by us at the Moscow Center in 2009. Despite the fact that his percentage of cells with incomplete chromatin condensation was on the same level as donors, the development of all zygotes stopped at an early stage. We studied of the centriole structure and reported the results of our work at a conference in Barcelona (Alieva et al., 2011). We had to wait for the second similar patient for more than 5 years after this (patient 1 in this paper from Tours Reproduction center). Initially, an ultrastructural study of his sperm was not conducted. It turned out that this patient had both types of developmental disorders — an increased proportion of spermatozoa with incomplete chromatin condensation and impaired disassembly of the centriolar adjunct.
Thus, it is not possible to obtain a larger number of patients for detailed research for two objective reasons: 1 - there are few such patients; 2- most of those who have similar problems do not try to establish the true cause of male sterility (they do not pay ultrastructural study), but immediately use sperm donor services.
It seems to us that not only the data from patients with IMS, but also the comparison of spermatozoa of healthy fertile donors and the fertile boar given in this article can be useful for understanding the role of a centriolar adjunct. Although there is conflicting evidence in the literature about disassembling a centriolar adjunct in human spermatozoa (the diagrams or photos show a human proximal centriole both with [3, 18] or without adjunct [2, 13], this process never was quantified even for healthy fertile donors.
Regarding the comparison of the centriolar adjunct and primary cilia, we believe that this is one of the main observations obtained as a result of our work in recent years. Two systems of terminology (proximal and distal centriole in sperm) and (mother and daughter centriole in somatic cells) existed in parallel for many years. Now it is possible to correlate these two terminologies.
For the first time, we received direct evidence that the sperm flagella grow from a daughter, and not from a mother centriole (Alieva et al. 2018), and answered one of the key questions that worried cell biologists. This is what gave us reason to call the centriolar adjunct cousin of primary cilia. It is, of course, not a twin or even not a brother, but his (her) cousin in our opinion. This allows us to extrapolate recent results on the role of primary cilia in the regulation of mitosis to the centriolar adjunct.
In our opinion, these results are important and interesting not only for specialists in reproduction biology, but also for a wide range of cellular biologists, which gave us reason to submit our article in the journal Cells, which has a wider readership than specialized reproductive journals.
Please see below for the comments.
Additional remarks:
Title
A statement in the title: “The centriolar adjunct – forgotten cousin of the primary cilium” is rather far-fetched and might be misleading. This manuscript does not provide any experimental data indicating that a centriolar adjunct and primary cilia are analogous organelles.
AR : What types of experiments can confirm or deny this? In this case, we proceed from the structure (absence of central MTs), the absence of motor function and connection with the mother centriole. The last principal statement was confirmed by us during the study of spermatids at various stages of development in our previous work (Alieva et al., 2018). The principal figure from this publication is available at this link:
https://www.researchgate.net/publication/327646211_Chapter_5_A_QUESTION_OF_FLAGELLA_ORIGIN_FOR_SPERMATIDS_-_MOTHER_OR_DAUGHTER_CENTRIOLE
From presented TEM photos of spermatids on Figure 10 (during later stages of spermiogenesis, the centrioles shift relative to each other, and at the earlier stage flagella is absent) it is obvious that the flagellum of the cilia grows from the daughter centriole (proximal end of daughter centriole is associated with the lateral surface of mother centriole). We also see in the photograph anlage of capitulum associated with the proximal (mother) centriole and anlage of segmented columns, associated with the distal (daughter) centriole. Consequently, the distal centriole giving rise to flagella is a daughter centriole, and the proximal centriole giving rise to the centriolar adjunct is the mother centriole.
A centriolar adjunct lacks central microtubules like primary cilia but in contrast to primary cilia a centriolar adjunct has triplet microtubules, B and C tubules are incomplete – but these are triplets not doublets (similar as in centriole or basal body - that also lack central microtubules). Moreover, in contrast to primary cilia, MTs of centriolar adjunct are surrounded by the electron-dense material. As I wrote in one of my comments below, if indeed a presence of the CA block zygotic division, there might be a functional analogy between primary cilium and CA. So, I agree with a cautions statement that “the CA ..might be partial analogous to another centriole derivate, the primary cilium”.
The expression used in the proposed title: “forgotten cousin of the primary cilium” is very eye-catching. I understand why it was used. But seeing such a title I would expect to find experimental data supporting such claim, not only the hypothesis. Therefore, I suggest to water down the title. e.g.
The centriolar adjunct - appearance and resorption in spermiogenesis and the potential impact on fertility”
Out of my natural scientific curiosity and in relation to the primary cilium. I wonder if microtubules of the centriolar adjunct are posttranslationally modified? (immunofluorescence, immunogold labelling: tubulin acetylation, detyrosination, glutamylation – antibodies are commercially available; microtubules in cilia, more specifically B-tubules, are highly acetylated, detyrosinated and glutamylated).
Also antibodies against some ciliary transition zone proteins are commercially available. Perhaps epitope targeted by the antibody is conserved and antibodies produced against mouse or human protein can be used.
Any IFT proteins near centriolar adjunct? Can these microtubules serve as a railway in some sort of transport?
If I am not mistaken it should be resorption (with “p”)
AR : “resorption” is reabsorption
“resorbtion” is progressive disassembly, disappearance
We thus used the correct definition of this process.
I am not a native speaker thus, I can be mistaken. However, I looked into the dictionaries and I could not find a noun: resorbtion, only a verb: to resorb. Also when describing a disassembly of the primary cilia, people use the word “resorption” e.g.
“The loss of primary cilia soon after the onset of Foxj1 expression also suggests that Foxj1 regulates a program for primary cilia resorption, possibly by promoting ...” Jain et al. Temporal relationship between primary and motile ciliogenesis in airway epithelial cells. Am J Respir Cell Mol Biol. 2010 Dec;43(6):731-9. doi: 10.1165/rcmb.2009-0328OC.
Merriam-Webster: Medical Definition of resorption
the action or process of resorbing something:.. age-related bone loss…is caused by a slight but persistent elevation in the rate of bone resorption over the rate of bone formation— P. S. Millard et al
I do not fill qualified to solve this issue, it is a question to a native speaker.
Abstract
“The average length of the “proximal centriole + centriolar adjunct” complex measured in sections, where it had parallel orientation in the section plane, was significantly longer in the spermatozoa of IMS patients than in the spermatozoa of healthy donors.”
Please change: “The average length….. was significantly longer…”
AR: If we understand correctly, you suggest to change the word "longer" to “greater”.
We’ve done it. (line 34)
Longer length does not seem right. I am not sure if greater or larger is correct, this is again a question to a native speaker.
Or simply you can state: We measured the “proximal centriole + centriolar adjunct” complex in sections, where it had parallel orientation. We found that on average proximal centriole + centriolar adjunct complex was significantly longer in the spermatozoa of IMS patients than in the spermatozoa of healthy donors.”
Introduction
lines 43-54
This fragment should be transferred to the discussion section. The content of the manuscript does not directly relate to the assisted reproductive technology but Authors hypothesis (that the centriolar adjunct length can be potentially used in the diagnostics) can be discussed.
AR: We agree with your recommendation and transfer this section to the end of the paper (lines 346-361). Additionally, we wrote small conclusion.
This paper gives an explanation of one of the reasons for stopping the development of a zygote after fertilization in vitro and transfers this case from the category of «idiopathic» to the category of diagnosable disorders of sperm development. (lines 361-364)
Thank you.
Lines 55-56:
“Several important morphological changes occur in spermatid maturation during last part of spermiogenesis during transformation into mature sperm cells.” – please rephrase.
“Several important morphological changes occur in spermatid maturation during last part of spermiogenesis during transformation their into mature sperm cells.”
AR: We added “their” in the sentence. (line 47)
This is obviously my mistake, sorry. It should be: Several important morphological changes occur in spermatid during last part of spermiogenesis, more precisely during their transformation into mature sperm cells.”
Line 93: “The aim of this study was to investigate the reason for infertility…” suggested a change: to investigate a cause …
AR: We agree with this comment and made a change in the text. (line 89)
Thank you.
Results
Fig. 1A – insets S1 and S2 should be bigger – this is a description of CA, so the structure of CA should be clearly visible.
AR : We agree with this remark and presented the inserts in the form of individual photos in modified figure 1. (lines 189-199)
Thank you.
Fig. 1B – the boundary between the proximal centriole and CA is well defined. What is the structure of this boundary? Does it resemble the structure of the transition zone?
AR: In Figure 1B, in the region of the boundary between the centriole and centriolar adjunct, the tangentially cut part of the first inner ring of the centriolar adjunct is visible (see the previous section in Photo 1a). In the lumen of the centriolar cylinder, irregularly dense granules were sometimes found (see Figs. 2a, 2d, 2g, 2j, 3a, 3e, 4b), but no additional structures such as the terminal plate were found.
Did you see serial cross-sections of this region? I believe that similar to primary cilia, such “transition zone” (here I mean a transitional region) has to be difficult to find in TEM. But have you seen any?
Fig. 1C – a dense material is visible at the level of the boundary between a proximal centriole and CA. Is this a collar-like structure? It is still visible in Fig. 1e
AR: It seems that «Сollar-like structure» is special terminus from another area. https://www.ncbi.nlm.nih.gov/pubmed/2417845
In this case, we think that it is part of striated columns that are adjacent to the proximal centriole from the outside.
Thank you. By a collar-like structure are meant something that surrounds like a collar. Please mark this structure on the image (also visible on image 1h? ) and add description to the Fig.1. so it will be of use to a broader audience.
New comment:
Fig. 1 description: Line 194: nucleus(a–c); three cross sections show…
Please add a space after a word “nucleus” (…nucleus (a-c)) In the description some letters indicating images are in regular font, some are in bold.
Fig. 2E – “…different number of asterisks means significant differences between the data groups”. Why P1 has two and P2, has three asterisks.
AR: A different number of stars between P1 and P2 shows that they are not only statistically different from donors, but also from each other. Centriolar adjunct in P1 patient spermatozoa usually longer then in spermatozoa of P1 patient.
I am sorry, but this is not clear. If in case of P1 patient centriolar adjunct is longer than in case of P2 patient, so the difference between control and P1 is bigger than between control and P2 (so why 2 stars in case of P1 and 3 stars in case of P2). I am confused. The difference between control and P1 is more significant and between control and P2. Alternatively, peoples used to draw a line indicating two groups of data that they wish to compare and mark starts above. In this way, Authors can indicate which samples / groups are compared: control versus P1, control versus P2 and P1 versus P2.
Fig. 2I – the 3D structure is not apparent on the presented reconstructed model of PC+CA in IMS patient. Please replace the image (maybe change the angle), please add reconstruction of PC+CA in a healthy individual.
AR: The aim of the 3D reconstruction was to show in a volumetric form the continuity of the MT of the proximal centriole and centriolar adjunct. The selected angle, moreover, shows optimally the separation of triplets and doublets into separate MTs at the end of the centriolar adjunct.
The reconstruction of the centriole of the sperm cell of the fertile donor was added by us as new Fig. S1 in two orientations with serial TEM cross sections.
Fig2l: 3D reconstruction of PC+CA complex in idiopathic male sterility (IMS) patient…
Please add: showing the continuity of the MT of the proximal centriole and centriolar adjunct.
I am not trying to be difficult. Maybe this is because of the pdf format but this continuity of the MT of the proximal centriole and centriolar adjunct is not well visible. Is it possible to mark each triplet with a different color?
Fig. 2D and K – a structure of the proximal centriole looks different in a healthy individual and IMS patient. Please comment.
AR: This small difference (angle of divergence of triplets) is caused by a difference in the level of the section. As is known (Vorobjev, Chentsov, 1982), the triplets at the proximal end of the centrioles are more “unfolded”. In cat spermatozoa this small difference was shown in the paper Sato and Oura, 1984 (Fig. 5).
Thank you.
Fig. 2A and G; Fig. 3B and D (and Supplementary Figs S1 and S2) – a proximal centriole in IMS patient seems to be longer. How often is it impossible to define a boundary between PC and CA? In case such boundary can be defined, is indeed PC longer in IMS patient?
AR: Unlike in the spermatids, the boundary between the centriolar adjunct and the proximal centriole cannot be accurately determined. That is why we measured the total length of the entire complex. As can be seen from the data on the length of centrioles in boar spermatozoa (Fig. 1d), the length of centrioles is not a strictly deterministic value, but has slight variations.
Thank you.
Lines 221-224: “Exploring the spermatozoa of healthy donors (as controls) and IMS patients, we found that, in some cells, the proximal centrioles were associated with structures similar to centriolar adjuncts, which we describe in porcine spermatids (Fig. 1 and 2). “– Please provide the % and N (number of all cells analyzed) of spermatozoa with CA (for healthy individuals and IMS patients).
AR: For each patient and donor, we have examined 2000–3000 spermatozoa. The percentage of centrioles with adjunct in patients was more than 90%, in donors about 50%. However, the data obtained on oblique sections do not make any sense, since they do not allow a quantitative analysis of the length of adjuncts. In addition, the oblique oriented centriole and adjunct lies on several successive sections, and it is impossible to determine at what distance in the volume of the last section the structure ends. Thus, situations "no adjunct" and "short adjunct" cannot be differ on oblique sections.
The sections parallel to the long axis of the complex make up only about 1 percent of all possible orientations in the object space. Thus, in order to find 30 suitable spermatozoa for analysis, we had to examine 2-3 thousand (two years of work!). This is exactly the case when§ in our opinion, a smaller but reliable sample provides much more information than a much larger, but not sufficiently reliable.
Thank you for an explanation. Can you include the information:” The percentage of centrioles with an adjunct in patients was more than 90%, in donors about 50%” in a main text?
Lines 224-226: “there was no electron-dense material in the walls and no discs in the internal volume of human spermatozoa centriolar adjuncts.” – If I am not mistaken this sentence describes a CA in mature human spermatozoa. CA with discs and electron dense material was observed in pig spermatids. Is this correct?
AR: You understood correctly. We could not examine human testicles (ethics requirements) therefore we’ve compared the structure of residual centriolar adjuncts in sperm of humans with spermatids in pigs. As far as we know, the centriolar adjunct has been specially studied in human spermatids only in one work De Kretser, 1969. Structure of centriolar adjunct in human spermatids in this work was practically identical to our photos from porcine spermatids.
Thank you for this information. Maybe you could mention De Kretser data in your text for clarity.
Don W Fawcett in the second edition (1981) of the book "The Cell" (Chapter 12: Centrioles) present centriolar adjunct in chinchilla spermatids (Fig. 314), which was less similar (internal lumen structure had not visible periodic structures) to a human one than porcine centriolar adjunct. On contrary, in the article of Kojima (1994) internal centriolar adjunct structure in goat (Fig. 3) was practical identical to porcine centriolar adjunct in our paper and human centriolar adjunct in De Kretser (1969, Fig. 11) paper.
Lines 230-231: “In some cases, the border between the centriolar adjunct (CA) and the proximal centriole (PC) was not sufficiently obvious” – please provide % and N of the analyzed cells
AR:We change phrase to: “The border between the centriolar adjunct (CA) and the proximal centriole (PC) was not obvious” (lines 210-211)
OK
Lines 267-268: “Fertility is proportional to the ratio of spermatozoa with a fully or almost completely disassembled centriolar adjunct (PC+CA complexes length class I).” – What is the base for this statement?
AR: We rewrite this paragraph:
All the spermatozoids of fertile boar had proximal centrioles with a completely disassembled centriolar adjunct (Fig. 1d, h-j). We can propose that fertilization capacity is proportional to the ratio of spermatozoa with a fully or almost completely disassembled centriolar adjunct (PC+CA complexes length class I). (lines 260-263)
Thank you.
Lines 269-270: “Indeed, boars with 100% fertility have proximal centrioles with a completely disassembled centriolar adjunct” – add REF. Do you mean male pig? Are there data showing that boars/ male pigs with CA has lower fertility? REF? How the fertility of male pigs was estimated?
AR: All the spermatozoids of fertile boar had proximal centrioles with a completely disassembled centriolar adjunct (Fig. 1d, h-j). We can have proposed that fertilization capacity is proportional to the ratio of spermatozoa with a fully or almost completely disassembled centriolar adjunct (PC+CA complexes length class I).
OK
Lines 278-282: “Based on the structure of the CA we propose that this structure might be partial analogous to another centriole derivate, the primary cilium.” Authors claim that based on: “CA does not have central MTs” and that “both primary cilia and CA are growing from the mother centriole”.
However, as Authors stated above, opposite to the ciliary axoneme:” The wall of the centriolar adjunct consists of electron-dense material with 9 complete MTs “A” and 18 incomplete MTs “B” and “C” immersed in it.” Presence of triplets contradicts the above hypothesis. In a very general way all structures formed on the base of centriole / basal body can be analog, not only primary cilium and CA.
The next Authors hypothesis (“These data allowed us to hypothesize that the CA may also be a regulator of first mitotic divisions, zygote cleavages”) is based on the first, yet unproven hypothesis that CA is an analogue of a primary cilium.
AR: We do not say that these structures are identical, like brothers or even twins; we say they look like cousins. This is really our hypothesis, which is based on growth from the mother centriole, the absence of central MTs and the absence of obvious motor function. Since we have a delayed division of zygotes associated with an unparsed centriolar adjunct, we draw a parallel with the delayed mitotic division of somatic cells when the primary cilia are not disassembled. The two hypotheses, when combined, give a logically substantiated confirmation of possible affinity, not only by origin (from the mother centriole), but also functionally by the primary cilia and the centriolar adjunct.
If indeed a presence of the CA blocks zygotic division, it might be a functional analogy. I agree.
Is it possible to investigate / observe changes in the proximal centriole when it regains a functionality during a zygotic division (staining with some antibodies against centrosomal proteins) and investigate if a presence of CA inhibits / delays this process in a model organism?
New comment:
Lines 276-279: “It thought sometimes, that distal centriole which forms the flagellum is mother centriole (like mother centriole which forms primary cilium in somatic cells); however, our data obtained on early porcine spermatids [23], showed that this is the daughter centriole that forms a flagellum.”
Please add “in sperm cells” for clarity
Lines 284-286: “These data allowed us to hypothesize that the CA can also be a regulator of first mitotic divisions,”
Suggestion: …CA can be a negative regulator..
Lines 290-293: “Malfunction of CA disassembly during spermiogenesis might indicate anomalies in regulatory kinase systems, and CA might serve as a kind of” brake” on a faulty embryonic process.” – Authors have no basis for this claim as kinase system was not investigated.
AR: This is only a hypothesis based on analogy with the data obtained for the primary cilia. Why not?
Well, why not.
Suggestion: One can hypothesize (one could speculate) that malfunction of CA disassembly during spermiogenesis could be caused by the anomalies in regulatory kinase systems, and CA might serve as a kind of” brake” on a faulty embryonic process.
New comment, lines 290-291:
In the first stage, disassembly of the primary cilium is regulated by AuroraA-HDAC6 kinase, while further disassembly depends on Plk1-Kif2A and Nek2-Kif24 kinases
Kinases are: Arora A, Plk1 and Nek2,
HDAC6 is deacetylase, Kif2A and Kif24 are motor proteins (kinesins)
Please correct the above statement. E.g. Phosphorylation of…by …kinase,
Lines 342-343:” Thus, we can conclude that disassembly of the centriolar adjunct is required for the functional maturity of human spermatozoa...”. However, as Authors indicated above: “…spermatozoa with condensed chromatin and disassembled (short) adjunct.” were present in patients (lines 323-328, statistics is missing). So there are other factors causing infertility.
AR: No doubt, you are right. Other factors and among them incomplete chromatin condensation have their effect on fertility. We have included in the Table 1 additional columns: "% of spermatozoa without centriolar adjunct (I class)" and “% of “normal spermatozoa” from two parameters: centriolar adjunct disassembly and chromatin condensation” and evaluated the decline in fertility in the aggregate of the two parameters studied for two patients and five donors. These two parameters, of course, do not completely cover all the factors affecting fertility, but they allow to estimate it approximately.
Abnormality of the various components of the flagellar axoneme occasionally may be noted in the spermatozoa of normal individuals (Zamboni, 1987, Pryor et al. 1981, Hunter and Deretzer 1986). The same can be said about other defects of sperm. Final fertility (or sterility) will depend on the totality of all disorders. For patients studied in our work only by two parameters, fertility decreased by 4-28 times in compare with fertile donors (who are not absolutely fertile too, like we see!). This led to the practical impossibility of developing normal embryos even when fertilized in vitro by pre-selected spermatozoids by morphological parameters.
Thank you.
I want to thank you for a detailed analysis of our work and constructive criticism, which, we dare to hope, has made our article more understandable and useful for readers.
Author Response
A centriolar adjunct lacks central microtubules like primary cilia but in contrast to primary cilia a centriolar adjunct has triplet microtubules, B and C tubules are incomplete – but these are triplets not doublets (similar as in centriole or basal body - that also lack central microtubules). Moreover, in contrast to primary cilia, MTs of centriolar adjunct are surrounded by the electron-dense material. As I wrote in one of my comments below, if indeed a presence of the CA block zygotic division, there might be a functional analogy between primary cilium and CA. So, I agree with a cautions statement that “the CA ..might be partial analogous to another centriole derivate, the primary cilium”.
The expression used in the proposed title: “forgotten cousin of the primary cilium” is very eye-catching. I understand why it was used. But seeing such a title I would expect to find experimental data supporting such claim, not only the hypothesis. Therefore, I suggest to water down the title. e.g.
I am not a native speaker thus, I can be mistaken. However, I looked into the dictionaries and I could not find a noun: resorbtion, only a verb: to resorb. Also when describing a disassembly of the primary cilia, people use the word “resorption” e.g.
“The loss of primary cilia soon after the onset of Foxj1 expression also suggests that Foxj1 regulates a program for primary cilia resorption, possibly by promoting ...” Jain et al. Temporal relationship between primary and motile ciliogenesis in airway epithelial cells. Am J Respir Cell Mol Biol. 2010 Dec;43(6):731-9. doi: 10.1165/rcmb.2009-0328OC.
Merriam-Webster: Medical Definition of resorption
the action or process of resorbing something: age-related bone loss…is caused by a slight but persistent elevation in the rate of bone resorption over the rate of bone formation— P. S. Millard et al
I do not fill qualified to solve this issue, it is a question to a native speaker.
Author response for second revision (AR 2): We agree to change our “eye-catching” title to more clear. Our colleague Dr. Michael Jennings, who is cell biologist and native English speaker recommended us to use the word “disassembly”:
The centriolar adjunct – appearance and disassembly in spermiogenesis and the potential impact on fertility.
The word “Resorbtion” was used by special company (Proof-Reading-Service.com Ltd (Devonshire, company registration number 8391405), which corrected English of first version of our paper.
Out of my natural scientific curiosity and in relation to the primary cilium. I wonder if microtubules of the centriolar adjunct are posttranslationally modified? (immunofluorescence, immunogold labelling: tubulin acetylation, detyrosination, glutamylation – antibodies are commercially available; microtubules in cilia, more specifically B-tubules, are highly acetylated, detyrosinated and glutamylated).
Also antibodies against some ciliary transition zone proteins are commercially available. Perhaps epitope targeted by the antibody is conserved and antibodies produced against mouse or human protein can be used.
Any IFT proteins near centriolar adjunct? Can these microtubules serve as a railway in some sort of transport?
AR 2: These are very interesting questions for which we have no answers yet. We think that MTs centriolar adjunct are acetylated, but for detailed research of other proteins we will need antibodies to specific porcine proteins.
Abstract
Longer length does not seem right. I am not sure if greater or larger is correct, this is again a question to a native speaker.
Or simply you can state: We measured the “proximal centriole + centriolar adjunct” complex in sections, where it had parallel orientation. We found that on average proximal centriole + centriolar adjunct complex was significantly longer in the spermatozoa of IMS patients than in the spermatozoa of healthy donors.”
AR 2: We agreed with this remark and wrote this sentence in the form close to that proposed by the reviewer, given the size of the abstract (200 words).
“We measured the average length of the “proximal centriole + centriolar adjunct” complex in sections, where it had parallel orientation in the section plane, and found that it was significantly longer in the spermatozoa of IMS patients than in the spermatozoa of healthy donors”.
Lines 55-56:
“Several important morphological changes occur in spermatid maturation during last part of spermiogenesis during transformation into mature sperm cells.” – please rephrase.
“Several important morphological changes occur in spermatid maturation during last part of spermiogenesis during transformation their into mature sperm cells.”
AR: We added “their” in the sentence. (line 47)
This is obviously my mistake, sorry. It should be: Several important morphological changes occur in spermatid during last part of spermiogenesis, more precisely during their transformation into mature sperm cells.”
AR 2: We agreed with this remark and wrote this sentence in the form that proposed by the reviewer.
Fig. 1B – the boundary between the proximal centriole and CA is well defined. What is the structure of this boundary? Does it resemble the structure of the transition zone?
AR: In Figure 1B, in the region of the boundary between the centriole and centriolar adjunct, the tangentially cut part of the first inner ring of the centriolar adjunct is visible (see the previous section in Photo 1a). In the lumen of the centriolar cylinder, irregularly dense granules were sometimes found (see Figs. 2a, 2d, 2g, 2j, 3a, 3e, 4b), but no additional structures such as the terminal plate were found.
Did you see serial cross-sections of this region? I believe that similar to primary cilia, such “transition zone” (here I mean a transitional region) has to be difficult to find in TEM. But have you seen any?
AR 2: I carefully looked again the photos of this series in the electronic archive (815 photos of porcine spermatids) and I found that photos 1e and 1f are two consecutive sections of one complete series (this is the “most cross” series) through the proximal centriole and the beginning of the centriolar adjunct. Therefore, between the last section of centriole (Fig. 1e) and the first section of the centriolar adjunct (Fig. 1f), there is no morphologically identifiable transition zone.
I shifted in Figure 1a indication of the location of the cross-sections in order to better fit the real situation.
Fig. 1C – a dense material is visible at the level of the boundary between a proximal centriole and CA. Is this a collar-like structure? It is still visible in Fig. 1e
AR: It seems that «Сollar-like structure» is special terminus from another area. https://www.ncbi.nlm.nih.gov/pubmed/2417845
In this case, we think that it is part of striated columns that are adjacent to the proximal centriole from the outside.
Thank you. By a collar-like structure are meant something that surrounds like a collar. Please mark this structure on the image (also visible on image 1h?) and add description to the Fig.1. so it will be of use to a broader audience.
AR 2: OK. We mark this structure on the image 1c, 1e, 1h and added the description to the Fig.1 legend.
New comment:
Fig. 1 description: Line 194: nucleus(a–c); three cross sections show…
Please add a space after a word “nucleus” (…nucleus (a-c)) In the description some letters indicating images are in regular font, some are in bold.
AR 2: Thanks, it's done.
Fig. 2E – “…different number of asterisks means significant differences between the data groups”. Why P1 has two and P2, has three asterisks.
AR: A different number of stars between P1 and P2 shows that they are not only statistically different from donors, but also from each other. Centriolar adjunct in P1 patient spermatozoa usually longer then in spermatozoa of P1 patient.
I am sorry, but this is not clear. If in case of P1 patient centriolar adjunct is longer than in case of P2 patient, so the difference between control and P1 is bigger than between control and P2 (so why 2 stars in case of P1 and 3 stars in case of P2). I am confused. The difference between control and P1 is more significant and between control and P2. Alternatively, peoples used to draw a line indicating two groups of data that they wish to compare and mark starts above. In this way, Authors can indicate which samples / groups are compared: control versus P1, control versus P2 and P1 versus P2.
AR 2: In order not to confuse the reader with the number of asterisks, we replaced them with a more usual designation - letters. Different letters indicate statistically significant differences between groups of data. In all case, the reader will find more complete information in the text of the paper.
Fig. 2I – the 3D structure is not apparent on the presented reconstructed model of PC+CA in IMS patient. Please replace the image (maybe change the angle), please add reconstruction of PC+CA in a healthy individual.
AR: The aim of the 3D reconstruction was to show in a volumetric form the continuity of the MT of the proximal centriole and centriolar adjunct. The selected angle, moreover, shows optimally the separation of triplets and doublets into separate MTs at the end of the centriolar adjunct.
The reconstruction of the centriole of the sperm cell of the fertile donor was added by us as new Fig. S1 in two orientations with serial TEM cross sections.
Fig2l: 3D reconstruction of PC+CA complex in idiopathic male sterility (IMS) patient…
Please add: showing the continuity of the MT of the proximal centriole and centriolar adjunct.
I am not trying to be difficult. Maybe this is because of the pdf format but this continuity of the MT of the proximal centriole and centriolar adjunct is not well visible. Is it possible to mark each triplet with a different color?
AR 2: We added this explanatory caption to the Figure 2 legend. Excuse me, but it will be difficult to modified our 3D reconstruction now.
Lines 221-224: “Exploring the spermatozoa of healthy donors (as controls) and IMS patients, we found that, in some cells, the proximal centrioles were associated with structures similar to centriolar adjuncts, which we describe in porcine spermatids (Fig. 1 and 2). “– Please provide the % and N (number of all cells analyzed) of spermatozoa with CA (for healthy individuals and IMS patients).
AR: For each patient and donor, we have examined 2000–3000 spermatozoa. The percentage of centrioles with adjunct in patients was more than 90%, in donors about 50%. However, the data obtained on oblique sections do not make any sense, since they do not allow a quantitative analysis of the length of adjuncts. In addition, the oblique oriented centriole and adjunct lies on several successive sections, and it is impossible to determine at what distance in the volume of the last section the structure ends. Thus, situations "no adjunct" and "short adjunct" cannot be differ on oblique sections.
The sections parallel to the long axis of the complex make up only about 1 percent of all possible orientations in the object space. Thus, in order to find 30 suitable spermatozoa for analysis, we had to examine 2-3 thousand (two years of work!). This is exactly the case when§ in our opinion, a smaller but reliable sample provides much more information than a much larger, but not sufficiently reliable.
Thank you for an explanation. Can you include the information:” The percentage of centrioles with an adjunct in patients was more than 90%, in donors about 50%” in a main text?
AR 2: Thanks, it's done. We added it in the text: “The percentage of centrioles with well visible centriolar adjunct in spermatozoa of both patients was more than 90%, in spermatozoa of all five donors about 50%”.
Lines 224-226: “there was no electron-dense material in the walls and no discs in the internal volume of human spermatozoa centriolar adjuncts.” – If I am not mistaken this sentence describes a CA in mature human spermatozoa. CA with discs and electron dense material was observed in pig spermatids. Is this correct?
AR: You understood correctly. We could not examine human testicles (ethics requirements) therefore we’ve compared the structure of residual centriolar adjuncts in sperm of humans with spermatids in pigs. As far as we know, the centriolar adjunct has been specially studied in human spermatids only in one work De Kretser, 1969. Structure of centriolar adjunct in human spermatids in this work was practically identical to our photos from porcine spermatids.
Thank you for this information. Maybe you could mention De Kretser data in your text for clarity.
Don W Fawcett in the second edition (1981) of the book "The Cell" (Chapter 12: Centrioles) present centriolar adjunct in chinchilla spermatids (Fig. 314), which was less similar (internal lumen structure had not visible periodic structures) to a human one than porcine centriolar adjunct.
On contrary, in the article of Kojima (1994) internal centriolar adjunct structure in goat (Fig. 3) was practical identical to porcine centriolar adjunct in our paper and human centriolar adjunct in De Kretser (1969, Fig. 11) paper.
AR 2: Thanks, it's done. We added it in the text: “So, the internal structure of the centriolar adjunct described by us in porcine spermatids was almost identical to the previously described structure of the centriolar adjunct in human spermatids [18].”
Lines 278-282: “Based on the structure of the CA we propose that this structure might be partial analogous to another centriole derivate, the primary cilium.” Authors claim that based on: “CA does not have central MTs” and that “both primary cilia and CA are growing from the mother centriole”.
However, as Authors stated above, opposite to the ciliary axoneme:” The wall of the centriolar adjunct consists of electron-dense material with 9 complete MTs “A” and 18 incomplete MTs “B” and “C” immersed in it.” Presence of triplets contradicts the above hypothesis. In a very general way all structures formed on the base of centriole / basal body can be analog, not only primary cilium and CA.
The next Authors hypothesis (“These data allowed us to hypothesize that the CA may also be a regulator of first mitotic divisions, zygote cleavages”) is based on the first, yet unproven hypothesis that CA is an analogue of a primary cilium.
AR: We do not say that these structures are identical, like brothers or even twins; we say they look like cousins. This is really our hypothesis, which is based on growth from the mother centriole, the absence of central MTs and the absence of obvious motor function. Since we have a delayed division of zygotes associated with an unparsed centriolar adjunct, we draw a parallel with the delayed mitotic division of somatic cells when the primary cilia are not disassembled. The two hypotheses, when combined, give a logically substantiated confirmation of possible affinity, not only by origin (from the mother centriole), but also functionally by the primary cilia and the centriolar adjunct.
If indeed a presence of the CA blocks zygotic division, it might be a functional analogy. I agree.
Is it possible to investigate / observe changes in the proximal centriole when it regains a functionality during a zygotic division (staining with some antibodies against centrosomal proteins) and investigate if a presence of CA inhibits / delays this process in a model organism?
AR 2: The main "problem" is that such an anomaly is found only in humans (there is data about very rarely in other primates). The ideal option would be to investigate the stopped embryos in patients with IMS. But they are destroyed (instead of investigating and assisting in the diagnosis) and their research is prohibited by law.
New comment:
Lines 276-279: “It thought sometimes, that distal centriole which forms the flagellum is mother centriole (like mother centriole which forms primary cilium in somatic cells); however, our data obtained on early porcine spermatids [23], showed that this is the daughter centriole that forms a flagellum.”
Please add “in sperm cells” for clarity
AR 2: Thanks, it's done.
Lines 284-286: “These data allowed us to hypothesize that the CA can also be a regulator of first mitotic divisions,”
Suggestion: …CA can be a negative regulator..
AR 2: Thanks, it's done.
Lines 290-293: “Malfunction of CA disassembly during spermiogenesis might indicate anomalies in regulatory kinase systems, and CA might serve as a kind of” brake” on a faulty embryonic process.” – Authors have no basis for this claim as kinase system was not investigated.
AR: This is only a hypothesis based on analogy with the data obtained for the primary cilia. Why not?
Well, why not.
Suggestion: One can hypothesize (one could speculate) that malfunction of CA disassembly during spermiogenesis could be caused by the anomalies in regulatory kinase systems, and CA might serve as a kind of” brake” on a faulty embryonic process.
One can hypothesize that malfunction of CA disassembly during spermiogenesis could be caused by the anomalies in regulatory kinase systems, and CA might serve as a kind of” brake” on a faulty embryonic process.
AR 2: OK, we replaced this sentence by phrase proposed by reviewer.
“One can hypothesize that malfunction of CA disassembly during spermiogenesis could be caused by the anomalies in regulatory kinase systems, and CA might serve as a kind of” brake” on a faulty embryonic process.”
New comment, lines 290-291:
In the first stage, disassembly of the primary cilium is regulated by AuroraA-HDAC6 kinase, while further disassembly depends on Plk1-Kif2A and Nek2-Kif24 kinases
Kinases are: Arora A, Plk1 and Nek2,
HDAC6 is deacetylase, Kif2A and Kif24 are motor proteins (kinesins)
Please correct the above statement. E.g. Phosphorylation of…by …kinase,
AR 2: Thanks. Here we directly use text from the paper, where kinases were together with substrates. In new version we used only the names of kinases:
“In the first stage, disassembly of the primary cilium is regulated by AuroraA kinase, while further disassembly depends on Plk1 and Nek2 kinases [24].”
First version of the paper was checked by Proof-Reading-Service.com Ltd (Devonshire, company registration number 8391405) and final version after second revision was corrected by Dr. Michael Jennings, who is cell biologist and native English speaker (green font color in the text).
I want to thank you for a detailed analysis of our work and constructive criticism, which, we dare to hope, has made our article more understandable and useful for readers.